# SyncTweedies: A General Generative Framework Based on Synchronized Diffusions

**Jaihoon Kim**[*]    **Juil Koo**[*]    **Kyeongmin Yeo**[*]    **Minhyuk Sung**

KAIST

{jh27kim,63days,aaaaa,mhsung}@kaist.ac.kr

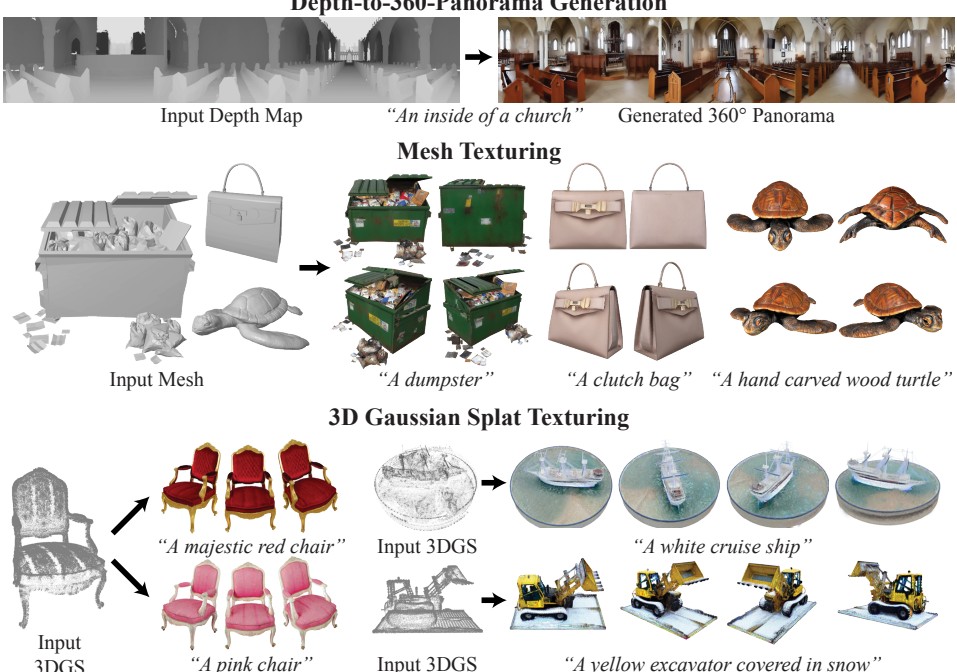

**Depth-to-360-Panorama Generation**

Input Depth Map    *"An inside of a church"*    Generated 360° Panorama

**Mesh Texturing**

Input Mesh    *"A dumpster"*    *"A clutch bag"*    *"A hand carved wood turtle"*

**3D Gaussian Splat Texturing**

*"A majestic red chair"*    Input 3DGS    *"A white cruise ship"*

Input 3DGS    *"A pink chair"*    Input 3DGS    *"A yellow excavator covered in snow"*

Figure 1: **Diverse visual content generated by** SyncTweedies**:** A diffusion synchronization process applicable to various downstream tasks without finetuning.

## Abstract

We introduce a general diffusion synchronization framework for generating diverse visual content, including ambiguous images, panorama images, 3D mesh textures, and 3D Gaussian splats textures, using a pretrained image diffusion model. We first present an analysis of various scenarios for synchronizing multiple diffusion processes through a canonical space. Based on the analysis, we introduce a synchronized diffusion method, SyncTweedies, which averages the outputs of Tweedie's formula while conducting denoising in multiple instance spaces. Compared to previous work that achieves synchronization through finetuning, SyncTweedies is a zero-shot method that does not require any finetuning, preserving the rich prior of diffusion models trained on Internet-scale image datasets without overfitting to specific domains. We verify that SyncTweedies offers the broadest applicability to diverse applications and superior performance compared to the previous state-of-the-art for each application. Our project page is at https://synctweedies.github.io.

---

[*]Equal contribution.

38th Conference on Neural Information Processing Systems (NeurIPS 2024).

# 1 Introduction

Image diffusion models [47, 38] have shown unprecedented ability to generate plausible images that are indistinguishable from real ones. The generative power of these models stems not only from their capacity to learn from a vast diversity of potential data but also from being trained on Internet-scale image datasets [49, 51].

Our goal is to expand the capabilities of pretrained image diffusion models to produce a wide range of 2D and 3D visual content, including panoramic images and textures for 3D objects, as shown in Figure 1, without the need to train diffusion models for each specific visual content. Despite the existence of general image datasets on the scale of billions [49], collecting other forms of visual data at this scale is not feasible. Nonetheless, most visual content can be converted into a regular image of a specific size through certain mappings, such as projecting for panoramic images and rendering for textures of 3D objects. Thus, we employ such a *bridging* function between each type of visual content and images, along with pretrained image diffusion models [47, 38].

We introduce a general generative framework that generates data points in the desired visual content space—referred to as canonical space—by combining the denoising process of diffusion models in the conventional image space—referred to as instance spaces. Given the bridging functions connecting the canonical space and instance spaces, we first explore performing individual denoising processes in each instance space while *synchronizing* them in the canonical space via the mapping. Another approach is to denoise directly in the canonical space, although it is not immediately feasible due to the absence of diffusion models trained on the canonical space. We investigate *redirecting* the noise prediction to the instance spaces but aggregating the outputs later in the canonical space.

Depending on the timing of aggregating the outputs of computation in the instance spaces, we identify *five* main possible options for the diffusion synchronization processes. Previous works [5, 18, 35] have investigated each of the possible cases only for specific applications, and none of them have analyzed and compared them across a range of applications. For the first time, we present a general framework for diffusion synchronization processes, within which the previous works [5, 18, 35] are contextualized as specific cases. We then present extensive analyses of different choices of diffusion synchronization processes. Based on the analyses, we demonstrate that the approach, which conducts denoising processes in *instance* spaces (not the canonical space) and synchronizes the outputs of Tweedie's formula [46] in the canonical space, provides the broadest applicability across a range of applications and the best performance. We name this approach `SyncTweedies` and showcase its superior performance in multiple visual content creation tasks compared with previous state-of-the-art methods.

Previous works [56, 34, 52, 63] finetune pretrained diffusion models to generate new types of outputs such as $360°$ panorama images and 3D mesh texture images. However, this approach requires a large quantity of target content for high-quality outputs which is prohibitively expensive to acquire. When it comes to generating visual content that can be parameterized into an image, a notable zero-shot approach not utilizing diffusion synchronization is Score Distillation Sampling (SDS) [41], which has shown particular effectiveness in 3D generation and texturing [31, 60, 62, 37]. However, this alternative application of diffusion models has been observed to produce suboptimal results and also requires a high CFG [22] weight for convergence, leading to over-saturation. For 3D mesh texture generation, specifically, an approach that iteratively updates each view image has also been explored in multiple previous works [10, 44, 8, 23, 17]. However, the accumulation of errors over iterations has been identified as a challenge. We demonstrate that our diffusion-synchronization-based approach outperforms these methods in terms of generation quality across various applications.

Overall, our contributions can be summarized as follows:

- We propose, for the first time, a general generative framework for diffusion synchronization processes.
- Through extensive analyses of various options for diffusion synchronization processes, including previous works [35, 18, 5, 33], we identify that the approach which synchronizes the outputs of Tweedie's formula and performs denoising in the instance space, `SyncTweedies`, offers the broadest applicability and superior performance.
- In our experiments, we verify the superior performance and versatility of `SyncTweedies` across diverse applications, including texturing on 3D meshes and Gaussian Splats [26], and depth-to-360-panorama generation. Compared to the previous state-of-the-art methods based on finetuning, optimization, and iterative updates, `SyncTweedies` demonstrates significantly better results.

## 2 Problem Definition

We consider a generative process that samples data within a space we term the *canonical* space $\mathcal{Z}$, where a pretrained diffusion model is not provided. Instead, we leverage diffusion models trained in other spaces called the *instance* spaces $\{\mathcal{W}_i\}_{i=1:N}$, where a *subset* of the canonical space can be instantiated into each of them via a mapping: $f_i : \mathcal{Z} \to \mathcal{W}_i$; we refer to this mapping as the *projection*. Let $g_i$ denote the *unprojection*, which is the inverse of $f_i$, mapping the instance space to a subset of the canonical space. We assume that the entire canonical space $\mathcal{Z}$ can be expressed as a composition of multiple instance spaces $\mathcal{W}_i$, meaning that for any data point $\mathbf{z} \in \mathcal{Z}$, there exist $\{\mathbf{w}_i \,|\, \mathbf{w}_i \in \mathcal{W}_i\}_{i=1:N}$ such that

$$\mathbf{z} = \mathcal{A}\left(\{g_i(\mathbf{w}_i)\}_{i=1:N}\right), \tag{1}$$

where $\mathcal{A}$ is an aggregation function that averages the data points from the multiple instance spaces in the canonical space. Our objective is to introduce a general framework for the generative process in the canonical space by integrating multiple denoising processes from different instance spaces through synchronization.

## 3 Diffusion Synchronization

We first outline the denoising procedure of DDIM [53] and then present possible options for diffusion synchronization processes based on it.

### 3.1 Denoising Process of DDIM [53]

Song *et al.* [53] have proposed DDIM, a generalized denoising process that controls the level of randomness during denoising. In DDIM [53], the posterior of the forward process is represented as follows:

$$q_{\sigma_t}\left(\mathbf{x}^{(t-1)}|\mathbf{x}^{(t)}, \mathbf{x}^{(0)}\right) = \mathcal{N}\left(\psi_{\sigma_t}^{(t)}(\mathbf{x}^{(t)}, \mathbf{x}^{(0)}), \sigma_t^2 \mathbf{I}\right), \tag{2}$$

where $\psi_{\sigma_t}^{(t)}(\mathbf{x}^{(t)}, \mathbf{x}^{(0)}) = \sqrt{\alpha_{t-1}}\mathbf{x}^{(0)} + \sqrt{\frac{1-\alpha_{t-1}-\sigma_t^2}{1-\alpha_t}} \cdot (\mathbf{x}^{(t)} - \sqrt{\alpha_t}\mathbf{x}^{(0)})$ and $\sigma_t$ is a hyperparameter determining the level of randomness. In this paper, we consider a deterministic process where $\sigma_t = 0$ for all $t$, thus $\psi_{\sigma_t=0}^{(t)}$ will be denoted as $\psi^{(t)}$ for simplicity. During denoising process, to sample $\mathbf{x}^{(t-1)}$ from its unknown original clean data point $\mathbf{x}^{(0)}$, we estimate $\mathbf{x}^{(0)}$ using Tweedie's formula [46]:

$$\mathbf{x}^{(0)} \simeq \phi^{(t)}(\mathbf{x}^{(t)}, \boldsymbol{\epsilon}_\theta(\mathbf{x}^{(t)})) = \frac{\mathbf{x}^{(t)} - \sqrt{1-\alpha_t}\boldsymbol{\epsilon}_\theta(\mathbf{x}^{(t)})}{\sqrt{\alpha_t}}, \tag{3}$$

where $\boldsymbol{\epsilon}_\theta$ is a noise prediction network, and for simplicity, the time input and condition term in $\boldsymbol{\epsilon}_\theta$ are dropped. In short, each deterministic denoising step of DDIM [53] is expressed as follows:

$$\mathbf{x}^{(t-1)} = \psi^{(t)}(\mathbf{x}^{(t)}, \phi^{(t)}(\mathbf{x}^{(t)}, \boldsymbol{\epsilon}_\theta(\mathbf{x}^{(t)}))). \tag{4}$$

### 3.2 Diffusion Synchronization Processes

We now explore various scenarios of sampling $\mathbf{z} \in \mathcal{Z}$ by leveraging the composition of multiple denoising processes in the instance spaces $\{\mathcal{W}_i\}_{i=1:N}$. Consider the denoising step of the diffusion model at each time step $t$ in each instance space $\mathcal{W}_i$:

$$\mathbf{w}_i^{(t-1)} = \psi^{(t)}(\mathbf{w}_i^{(t)}, \phi^{(t)}(\mathbf{w}_i^{(t)}, \boldsymbol{\epsilon}_\theta(\mathbf{w}_i^{(t)}))). \tag{5}$$

A naïve approach to generating data in the canonical space through the denoising processes in instance spaces would be to perform the processes independently in each instance space and then aggregate the final denoised outputs in the canonical space at the end using the averaging function $\mathcal{A}$. However, this approach results in poor outcomes that lack consistency across outputs in different instance spaces. Hence, we propose to *synchronize* the denoising processes at each time step $t$ through the unprojection operation $g_i$ from each instance space to the canonical space and the aggregation operation $\mathcal{A}$, after which the results will be back-projected via the projection operation $f_i$ to each instance space again. Note that, as described in Equation 4, the estimated mean of the posterior distribution $\psi^{(t)}(\cdot, \cdot)$ involves multiple layers of computations: noise prediction $\boldsymbol{\epsilon}_\theta(\cdot)$, Tweedie's formula [46] $\phi^{(t)}(\cdot, \cdot)$ approximating the final output $\mathbf{x}^{(0)}$ each time step, and the final linear combination $\psi^{(t)}(\cdot, \cdot)$. Synchronization through the sequence of unprojection $g_i$, aggregation in the canonical space $\mathcal{A}$, and projection $f_i$ can thus be performed after each layer of these computations, resulting in the following three cases:

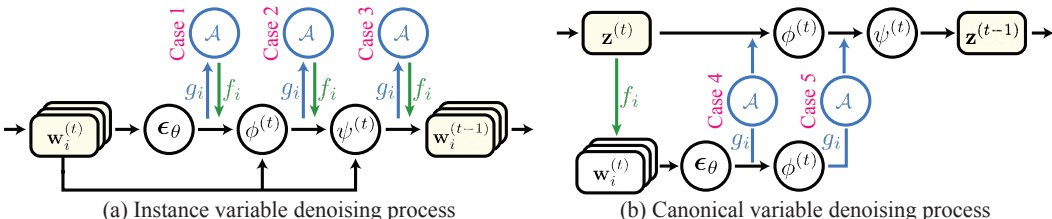

(a) Instance variable denoising process      (b) Canonical variable denoising process

Figure 2: **Diagrams of diffusion synchronization processes.** The left diagram depicts denoising instance variables $\{\mathbf{w}_i\}$, while the right diagram illustrates directly denoising a canonical variable $\mathbf{z}$.

$$\text{Case 1} : \mathbf{w}_i^{(t-1)} = \psi^{(t)}(\mathbf{w}_i^{(t)}, \phi^{(t)}(\mathbf{w}_i^{(t)}, f_i(\mathcal{A}(\{g_j(\boldsymbol{\epsilon}_\theta(\mathbf{w}_j^{(t)}))\}_{j=1}^N))))$$

$$\text{Case 2} : \mathbf{w}_i^{(t-1)} = \psi^{(t)}(\mathbf{w}_i^{(t)}, f_i(\mathcal{A}(\{g_j(\phi^{(t)}(\mathbf{w}_j^{(t)}, \boldsymbol{\epsilon}_\theta(\mathbf{w}_j^{(t)})))\}_{j=1}^N)))$$

$$\text{Case 3} : \mathbf{w}_i^{(t-1)} = f_i(\mathcal{A}(\{g_j(\psi^{(t)}(\mathbf{w}_j^{(t)}, \phi^{(t)}(\mathbf{w}_j^{(t)}, \boldsymbol{\epsilon}_\theta(\mathbf{w}_j^{(t)}))))\}_{j=1}^N)).$$

In each case, we highlight the computation layer to be synchronized in red.

Another notable approach is to conduct the denoising process directly on the canonical space:

$$\mathbf{z}^{(t-1)} = \psi^{(t)}(\mathbf{z}^{(t)}, \phi^{(t)}(\mathbf{z}^{(t)}, \boldsymbol{\epsilon}_\theta(\mathbf{z}^{(t)})))), \tag{6}$$

although it is not directly feasible because the noise prediction network in the canonical space $\boldsymbol{\epsilon}_\theta(\mathbf{z}^{(t)})$ is not available. Nevertheless, it can be achieved by *redirecting* the noise prediction to the instance spaces as follows:

(a) project the intermediate noisy data point $\mathbf{z}^{(t)}$ from the canonical space to each instance space, resulting in $f_i(\mathbf{z}^{(t)})$,

(b) apply a *subsequence* of the operations: $\boldsymbol{\epsilon}_\theta$, $\phi^{(t)}$, and $\psi^{(t)}$,

(c) unproject the outputs back to the canonical space via $g_i$ and then average them using the aggregation function $\mathcal{A}$, and

(d) perform the remaining operations in the canonical space.

Such an approach of performing the denoising process in the canonical space leads to the following two additional cases depending on the subsequence of operations at step (b):

$$\text{Case 4} : \mathbf{z}^{(t-1)} = \psi^{(t)}(\mathbf{z}^{(t)}, \phi^{(t)}(\mathbf{z}^{(t)}, \mathcal{A}(\{g_i(\boldsymbol{\epsilon}_\theta(f_i(\mathbf{z}^{(t)})))\}_{i=1}^N))))$$

$$\text{Case 5} : \mathbf{z}^{(t-1)} = \psi^{(t)}(\mathbf{z}^{(t)}, \mathcal{A}(\{g_i(\phi^{(t)}(f_i(\mathbf{z}^{(t)}), \boldsymbol{\epsilon}_\theta(f_i(\mathbf{z}^{(t)}))))\}_{i=1}^N)).$$

Illustration of the aforementioned diffusion synchronization processes are shown in Figure 2. Note the analogy between Cases 1 and 4, and Cases 2 and 5 in terms of the variable averaged in the canonical space with the aggregation operator $\mathcal{A}$: either the outputs of $\boldsymbol{\epsilon}_\theta(\cdot)$ or $\phi^{(t)}(\cdot, \cdot)$.

While it is also feasible to conduct the aggregation $\mathcal{A}$ multiple times with the output of different layers within a single denoising step, and to denoise data both in instance spaces and the canonical space, we empirically find that such more convoluted cases perform worse. In **Appendix H**, we detail our exploration of all possible cases and present experimental analyses.

### 3.3 Connection to Previous Diffusion Synchronization Methods

Below, we first review previous works each corresponding to a specific case of the aforementioned possible diffusion synchronization processes while focusing on a specific application. Then, we discuss finetuning-based approaches and their limitations. In Section 4, we also review literature targeting the same applications but without diffusion synchronization.

#### 3.3.1 Zero-Shot-Based Methods

**Ambiguous Image Generation.** Ambiguous images are images that exhibit different appearances under certain transformations, such as a $90°$ rotation or flipping. They can be generated through

a diffusion synchronization process, considering both the canonical space $\mathcal{Z}$ and instance spaces $\{\mathcal{W}_i\}_{i=1:N}$ as the same space of the image, with the projection operation $f_i$ representing the transformation producing each appearance. Visual Anagrams [18] uses Case 4 which aggregates the noise predictions $\epsilon_\theta(\cdot)$ to generate ambiguous images.

**Arbitrary-Sized Image Generation.** In arbitrary-sized image generation, the canonical space $\mathcal{Z}$ is the space of the arbitrary-sized image, while the instance spaces $\{\mathcal{W}_i\}_{i=1:N}$ are overlapping patches across the arbitrary-sized image, matching the resolution of the images that the pretrained image diffusion model can generate. The projection operation $f_i$ corresponds to the cropping operation applied to each patch. MultiDiffusion [5] and SyncDiffusion [29] introduce arbitrary-sized image generation methods using Case 3, averaging the mean of the posterior distribution $\psi^{(t)}(\cdot, \cdot)$.

**Mesh Texturing.** In 3D mesh texturing, the texture image space serves as the canonical space $\mathcal{Z}$, and the rendered images from each view serve as the instance spaces $\{\mathcal{W}_i\}_{i=1:N}$. The projection operation $f_i$ corresponds to rendering 3D textured meshes into 2D images. SyncMVD [35] proposes a 3D mesh texturing method by leveraging Case 5, which performs denoising in the canonical space and unprojects the outputs of Tweedie's formula [46] $\phi^{(t)}(\cdot, \cdot)$.

### 3.3.2 Finetuning-Based Methods

In addition to the aforementioned works, there have been attempts to achieve synchronization through finetuning. In multi-view image generation, SyncDreamer [34] and MVDream [52] finetune pretrained image diffusion models to achieve consistency across different views. MVDiffusion [56] and DiffCollage [65] generate $360°$ panorama images through finetuning. Additionally, Paint3D [63] trains an encoder to directly generate 3D mesh texture images in the UV space. However, these finetuning-based methods use target sample datasets [16, 9, 15, 13] that are smaller by *orders of magnitude* compared to Internet-scale image datasets [49], e.g., 10K panorama images [9] vs. 5B images [49]. As a result, they are prone to ovefitting and losing the rich prior and generalizability of pretrained image diffusion models [47, 48]. Additionally, the poor quality of textures in most 3D model datasets results in unsatisfactory texturing outcomes, even when using relatively large-scale datasets [16, 15]. In our experiments, we demonstrate that our zero-shot synchronization method, fully leveraging the pretrained model without bias toward a specific dataset, provides the best realism and widest diversity, assessed by FID and KID, compared to the finetuning-based methods.

Table 1: **A quantitative comparison in ambiguous image generation.** KID [6] is scaled by $10^3$. For each row, we highlight the column whose value is within 95% of the best.

| Projection | Metric | Case 1 | SyncTweedies Case 2 | Case 3 | Visual Anagrams [18] Case 4 | Case 5 |
|---|---|---|---|---|---|---|
| 1-to-1 Projection | CLIP-A [18] ↑ | 30.35 | 30.4 | 30.32 | 30.35 | 30.34 |
| | CLIP-C [18] ↑ | 64.52 | 64.48 | 64.49 | 64.59 | 64.48 |
| | FID [21] ↓ | 85.88 | 86.74 | 85.69 | 86.35 | 86.54 |
| | KID [6] ↓ | 32.37 | 32.59 | 32.57 | 32.41 | 32.86 |
| 1-to-$n$ Projection | CLIP-A [18] ↑ | 25.97 | 30.16 | 29.94 | 25.64 | 30.23 |
| | CLIP-C [18] ↑ | 54.77 | 60.86 | 60.64 | 54.15 | 61.01 |
| | FID [21] ↓ | 232.65 | 110.51 | 117.84 | 257.53 | 108.22 |
| | KID [6] ↓ | 216.71 | 77.16 | 85.52 | 257.43 | 74.48 |
| $n$-to-1 Projection | CLIP-A [18] ↑ | 21.28 | 29.56 | 21.58 | 21.33 | 21.09 |
| | CLIP-C [18] ↑ | 49.94 | 63.1 | 50.58 | 50.05 | 50.04 |
| | FID [21] ↓ | 405.82 | 96.3 | 243.23 | 301.2 | 289.82 |
| | KID [6] ↓ | 496.98 | 40.91 | 151.11 | 233.11 | 213.45 |

## 3.4 Comparison Across the Diffusion Synchronization Processes

Here, we compare the five cases of diffusion synchronization processes in Section 3.2 and analyze their characteristics through various toy experiments.

### 3.4.1 Toy Experiment Setup: Ambiguous Image Generation

For the toy experiment setup, we employ the task of generating ambiguous images introduced by Geng *et al.* [18] (see Section 3.3.1 for descriptions of ambiguous images). In this setup, we consider two-view ambiguous image generation, where two different transformations are applied, each producing a distinct appearance. Note that one of the transformations is an identity transformation, while the other is chosen to simulate different scenarios of mapping pixels from the canonical space

| Case 1 | | SyncTweedies
**Case 2** | | Case 3 | | Visual Anagrams [18]
Case 4 | | Case 5 | |
|---|---|---|---|---|---|---|---|---|---|
| $\mathbf{w}_1^{(0)}$ | $\mathbf{w}_2^{(0)}$ | $\mathbf{w}_1^{(0)}$ | $\mathbf{w}_2^{(0)}$ | $\mathbf{w}_1^{(0)}$ | $\mathbf{w}_2^{(0)}$ | $\mathbf{w}_1^{(0)}$ | $\mathbf{w}_2^{(0)}$ | $\mathbf{w}_1^{(0)}$ | $\mathbf{w}_2^{(0)}$ |

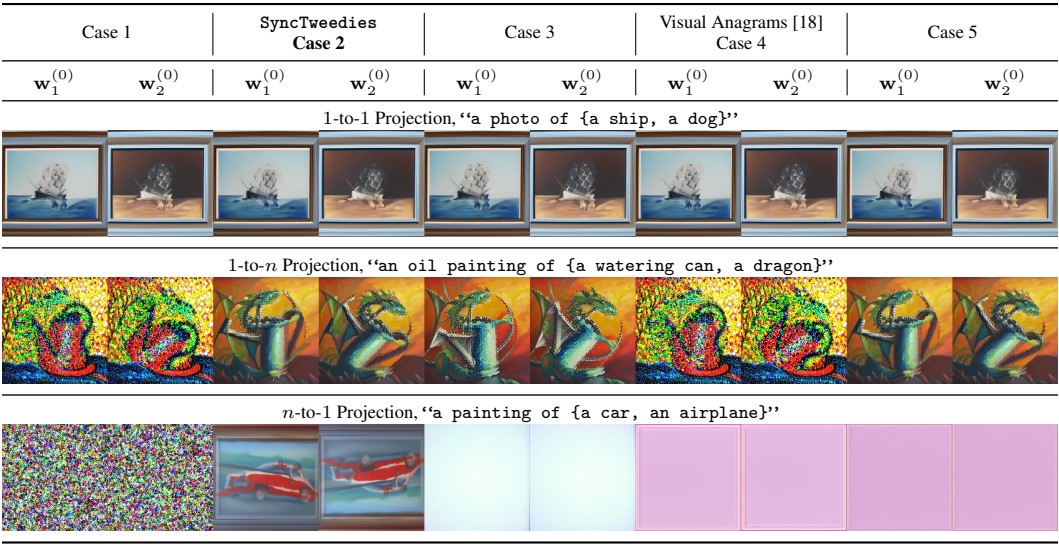

Figure 3: **Qualitative results of ambiguous image generation.** While all diffusion synchronization processes show identical results with 1-to-1 projections, Case 1, Case 3 and Visual Anagrams [18] (Case 4) exhibit degraded performance when the projections are 1-to-$n$. Notably, SyncTweedies can be applied to the widest range of projections, including $n$-to-1 projections, where Case 5 fails to generate plausible outputs.

to the instance space: 1-to-1, 1-to-$n$, and $n$-to-1 projection. In 1-to-1 and $n$-to-1 projections, we use the 10 transformations from Visual Anagrams [18], while for the 1-to-$n$ projection, we apply rotation transformations with randomly sampled angles. For all projection cases, we use the 95 prompts from [18]. For more details on the experiment setups, refer to **Appendix** B.1.

### 3.4.2   1-to-1 **Projection**

In 1-to-1 projection case, the five cases of diffusion synchronization processes become identical, as shown in **Appendix** D. The quantitative and qualitative results of diffusion synchronization processes are presented in Table 1 and the first row of Figure 3, respectively, where the fully denoised instance variables, $\mathbf{w}_1^{(0)}$ and $\mathbf{w}_2^{(0)}$, are displayed side by side. The results confirm that all diffusion synchronization processes produce the same outputs.

### 3.4.3   1-to-$n$ **Projection**

We further investigate the five cases of diffusion synchronization processes with different transformations for ambiguous images. It is important to note that all the transformations previously mentioned are perfectly invertible, meaning: $f_i(g_i(\mathbf{w}_i)) = \mathbf{w}_i$. However, in certain applications, the projection $f_i$ is often not a *function* but an 1-to-$n$ mapping, thus not allowing its inverse. For example, consider generating a texture image of a 3D object while treating the texture image space as the canonical space and the rendered image spaces as instance spaces. When mapping each pixel of a specific view image to a pixel in the texture image in the rendering process—with nearest neighbor sampling, one pixel in the texture space can be projected to multiple pixels. Hence, the unprojection $g_i$ cannot be a perfect inverse of the projection $f_i$ but can only be an approximation, making the reprojection error $\|\mathbf{w}_i - f_i(g_i(\mathbf{w}_i))\|$ small. This violates the initial conditions required for the proof in **Appendix** D that states Cases 1-5 become identical, and we observe that such a case of having 1-to-$n$ projection $f_i$ can significantly impact the diffusion synchronization process.

As a toy experiment setup illustrating such a case with ambiguous image generation, we replace the 1-to-1 transformations used in Section 3.4.2 to rotation transformations with nearest-neighbor sampling. We randomly select an angle and rotate an inner circle of the image while leaving the rest of the region unchanged. Due to discretization, rotating an image followed by an inverse rotation may not perfectly restore the original image.

The second row of Table 1 and Figure 3 present the quantitative and qualitative results of 1-to-$n$ projection experiment. Note that the performance of Case 1 and Visual Anagrams [18] (Case

4), which aggregate the predicted noises $\epsilon_\theta(\cdot)$ from either instance variables $\mathbf{w}_i^{(t)}$ or a projected canonical variable $f_i(\mathbf{z}^{(t)})$ respectively, significantly declines. Also, the performance of Case 3, which aggregates the posterior means $\psi^{(t)}(\cdot, \cdot)$, shows a minor decline. The quality of Cases 2 and 5, however, remain almost unchanged. This highlights that the denoising process is highly sensitive to the predicted noise and to the intermediate noisy data points, while it is much more robust to the outputs of Tweedie's formula [46] $\phi^{(t)}(\cdot, \cdot)$, the prediction of the final clean data point at an intermediate stage.

### 3.4.4 $n$-to-$1$ Projection

Then, do the results above conclude that both Cases 2 and 5 are suitable for all applications? Lastly, we consider the case when the projection $f_i$ also involves an $n$-to-1 mapping. Such a scenario can arise when coloring not a solid mesh but a neural 3D representation rendered with the volume rendering equation [25, 26, 39]. Due to the nature of volume rendering, which involves sampling *multiple* points along a ray and taking a weighted sum of their information, the projection operation $f_i$ includes an $n$-to-1 mapping. Note that this case also violates the initial conditions of the proof in **Appendix** D, which states that the diffusion synchronization cases become identical under specific initial conditions. Additionally, Case 5 results in poor outcomes due to a *variance decrease* issue. Let $\{\mathbf{x}_i\}_{i=1:N}$ be random variables, each sampled from $\mathbf{x}_i \sim \mathcal{N}(\boldsymbol{\mu}_i, \sigma_t^2 \mathbf{I})$, and $\mathbf{x} = \sum_{i=1}^N w_i \mathbf{x}_i$ be the weighted sum, where $0 \le w_i \le 1$ and $\sum_{i=1}^N w_i = 1$. Then, $\mathbf{x}$ also follows the Gaussian distribution $\mathbf{x} \sim \mathcal{N}\left(\sum_{i=1}^N w_i \boldsymbol{\mu}_i, \sum_{i=1}^N w_i^2 \sigma_t^2 \mathbf{I}\right)$. From the triangle inequality [40], the sum of squares is always less than or equal to the square of the sum: $\sum_{i=1}^N w_i^2 \le (\sum_{i=1}^N w_i)^2 = 1$, implying that the variance of $\mathbf{x}$ is mostly less than the variance of $\mathbf{x}_i$.

Consequently, when $f_i$ includes an $n$-to-1 mapping, the variance of $\mathbf{w}_i^{(t)}$, computed as a weighted sum over multiple points in the canonical space, is mostly less than the variance of $\mathbf{z}^{(t)}$. Thus, the final output of Case 5 becomes blurry and coarse since each intermediate noisy latent in instance spaces $\mathbf{w}_i^{(t)}$ experiences a decrease in variance compared to that of $\mathbf{z}^{(t)}$.

We validate our analysis with another toy experiment, where we use the same set of transformations used by Geng *et al.* [18] but with a multiplane image (MPI) [57] as the canonical space. The image of each instance space is rendered by first averaging colors in the multiplane of the canonical space and then applying the transformation. Ten planes are used for the multiplane image representation in our experiments. The results are presented in the third row of Table 1 and Figure 3. Notably, Case 5 fails to produce plausible images like the other cases, whereas Case 2 still generates realistic images.

Table 2 below summarizes suitable cases for each projection type. Note that Case 2 is the only case that is applicable to any type of projection function. Since Case 2 involves averaging the outputs of Tweedie's formula in the instance spaces, we name this case `SyncTweedies`. Experimental results with additional applications are demonstrated in Section 5, and analysis of all possible cases is presented in **Appendix** H.

Table 2: **Analysis of diffusion synchronization processes on different projection scenarios.** `SyncTweedies` offers the broadest range of applications.

| Projection | Application | Case 1 | SyncTweedies Case 2 | Case 3 | Case 4 | Case 5 |
|---|---|---|---|---|---|---|
| 1-to-1 | Ambiguous images, Arbitrary-sized images | ✔ | ✔ | ✔ | ✔ | ✔ |
| 1-to-$n$ | 360° panoramas, 3D mesh texturing | ✗ | ✔ | ✗ | ✗ | ✔ |
| $n$-to-1 | 3D Gaussian Splats [26] texturing | ✗ | ✔ | ✗ | ✗ | ✗ |
| | Previous Work | - | - | MultiDiffusion [5] | Visual Anagrams [18] | SyncMVD [35] |

## 4 Related Work

In addition to Section 3.3.1 introducing previous works on diffusion synchronization, in this section, we review other previous works that utilize pretrained diffusion models in different ways to generate or edit visual content.

**Optimization-Based Methods.** Poole *et al.* [41] first introduced Score Distillation Sampling (SDS), which facilitates data sampling in a canonical space by leveraging the loss function of the diffusion model training and performing gradient descent. This idea, originally introduced for 3D generation [58, 31, 55], has been widely applied to various applications, including vector image generation [24], ambiguous image generation [7], mesh texturing [37, 11, 62], mesh deformation [61], and 4D generation [32, 3]. Subsequent works [20, 27, 28] also proposed modified loss functions not to generate data but to edit existing data while preserving their identities. This approach, exploiting diffusion models not for denoising but for gradient-descent-based updating, generally produces less realistic outcomes and is more time-consuming compared to denoising-based generation.

**Iterative View Updating Methods.** Particularly for 3D object/scene texturing and editing, there are approaches to iteratively update each view image and subsequently refine the 3D object/scene. TEXTure [44], Text2Tex [10], and TexFusion [8] are previous works that sequentially update a partial texture image from each view and unproject it onto the 3D object mesh. For texturing 3D scene meshes, Text2Room [23] and SceneScape [17] take a similar approach and update scene textures sequentially. Instruct-NeRF2NeRF [19] proposed to edit a 3D scene by iteratively replacing each view image used in the reconstruction process. However, sequentially updating the canonical sample leads to error accumulations, resulting in blurriness or inconsistency across different views.

**Utilization of One-Step Predictions.** Previous works have utilized the outputs of Tweedie's formula to restore images [12, 66] and to guide the generation process [4, 29]. However, the one-step predicted samples are used to compute the gradient from a predefined loss function to guide the sampling process, rather than for synchronization, which differentiates from our approach.

Concurrent works [50, 59] also average the outputs of Tweedie's formula, similar to our approach, but they focus only on specific applications. For the first time, we present a general framework for diffusion synchronization and provide a comprehensive analysis of different synchronization methods across various applications.

Table 3: **A quantitative comparison in 3D mesh texturing.** KID is scaled by $10^3$. The best in each row is highlighted by **bold**.

| Metric | Diffusion Synchronization | | | | | Finetuning-Based | Optim.-Based | Iter. View Updating | |
| --- | --- | --- | --- | --- | --- | --- | --- | --- | --- |
| | Case 1 | Sync-Tweedies **Case 2** | Case 3 | Case 4 | Sync-MVD [35] Case 5 | Paint3D [63] | Paint-it [62] | TEXTure [44] | Text2Tex [10] |
| FID [21] ↓ | 135.61 | **21.76** | 36.12 | 131.67 | 22.76 | 31.66 | 28.23 | 34.98 | 26.10 |
| KID [6] ↓ | 68.63 | **1.46** | 6.60 | 65.70 | 1.74 | 5.69 | 2.30 | 6.83 | 2.51 |
| CLIP-S [42] ↑ | 25.26 | **28.89** | 27.88 | 25.31 | 28.82 | 28.04 | 28.55 | 28.63 | 27.94 |

| Prompt | Diffusion Synchronization | | | | | Finetuning-Based | Optim.-Based | Iter. View Updating | |
| --- | --- | --- | --- | --- | --- | --- | --- | --- | --- |
| | Case 1 | Sync-Tweedies **Case 2** | Case 3 | Case 4 | Sync-MVD [35] Case 5 | Paint3D [63] | Paint-it [62] | TEXTure [44] | Text2Tex [10] |
| ``Minivan'' | | | | | | | | | |
| ``Baseball glove'' | | | | | | | | | |
| ``Light bulb'' | | | | | | | | | |

Figure 4: **Qualitative results of 3D mesh texturing.** `SyncTweedies` and SyncMVD [35] generate realistic texture images, achieving better results than other baselines including finetuning-based method. Other diffusion synchronization cases fail to produce plausible textures.

## 5 Applications

We quantitatively and qualitatively compare `SyncTweedies` with the other diffusion synchronization processes, as well as the state-of-the-art methods of each application: 3D mesh texturing (Section 5.1), depth-to-360-panorama generation (Section 5.2) ,and 3D Gaussian splats [26] texturing (Section 5.3). Additional experiments and detailed setups are provided in **Appendix**, including (1) additional

Table 4: **A quantitative comparison in depth-to-360-panorama application.** KID is scaled by $10^3$. The best in each row is highlighted by **bold**.

| Metric | Case 1 | SyncTweedies Case 2 | Case 3 | Case 4 | Case 5 | MVDiffusion [56] |
|---|---|---|---|---|---|---|
| FID [21] ↓ | 364.61 | **42.11** | 55.95 | 348.18 | 43.39 | 80.51 |
| KID [6] ↓ | 375.42 | **21.19** | 34.67 | 362.77 | 22.87 | 56.91 |
| CLIP-S [42] ↑ | 19.75 | **28.01** | 27.19 | 19.93 | 27.99 | 24.74 |

qualitative results, (2) implementation details of each application, (3) arbitrary-sized image generation, (4) 3D mesh texture editing and diversity comparison, (6) runtime and VRAM usage comparisons, and (7) user preference evaluations.

**Experiment Setup.** In the case of instance variable denoising processes introduced in Section 3.2 (Cases 1-3), we initialize instance variables by projecting an initial canonical latent $\mathbf{z}^{(T)}$ sampled from a standard Gaussian distribution $\mathcal{N}(\mathbf{0}, \mathbf{I})$: $\mathbf{w}_i^{(T)} \leftarrow f_i(\mathbf{z}^{(T)})$. For $n$-to-1 projection cases (e.g.,3D Gaussian splats texturing), the instance variables are directly initialized from a standard Gaussian distribution which can avoid the variance decrease issue discussed in Section 3.4.4.

For instance space denoising processes, the final canonical variables are obtained by synchronizing the fully denoised instance variables at the end of the diffusion synchronization processes. Refer to Section 3.3.1 for the detailed definition of the canonical space $\mathcal{Z}$, the instance spaces $\{\mathcal{W}_i\}_{i=1:N}$, the projection operation $f_i$, and the unprojection operation $g_i$ in each application.

**Evaluation Setup.** Across all applications, we compute FID [21] and KID [6] to assess the fidelity of the generated images and CLIP similarity [42] (CLIP-S) to evaluate text alignment. We use a depth-conditioned ControlNet [64] as the pretrained image diffusion model.

## 5.1 3D Mesh Texturing

In 3D mesh texturing, projection operation $f_i$ is a rendering function which outputs perspective view images from a 3D mesh with a texture image. This operation represents a 1-to-$n$ projection due to discretization. We evaluate five diffusion synchronization cases along with Paint3D [63], a finetuning-based method, Paint-it [62], an optimization-based method, and TEXTure [44] and Text2Tex [10], which are iterative-view-updating-based methods. We use 429 pairs of meshes and prompts used in TEXTure [44] and Text2Tex [10].

**Results.** We present quantitative and qualitative results in Table 3 and Figure 4, respectively. The results in Table 3 align with the observations shown in the 1-to-$n$ projection case discussed in Section 3.4.4. SyncTweedies and SyncMVD [35] outperform other baselines across all metrics, but ours demonstrates superior performance compared to SyncMVD.

Notably, SyncTweedies outperforms Paint3D [63], a finetuning-based method, indicating that finetuning with a relatively small set of synthetic 3D objects [16] is not sufficient for realistic texture generation. This is further evidenced by the cartoonish texture of the car in row 1 of Figure 4. Optimization-based and iterative-view-updating-based methods produce unrealistic texture images, often exhibiting high saturation and visible seams, as seen in the baseball glove and light bulb in rows 2 and 3 of Figure 4. These issues are also reflected in the relatively high FID and KID scores in Table 3. See **Appendix** A for additional qualitative results.

## 5.2 Depth-to-360-Panorama

We generate $360°$ panorama images from $360°$ depth maps obtained from the 360MonoDepth [43] dataset. Here, $f_i$ projects a $360°$ panorama to a perspective view image, which is an 1-to-$n$ projection due to discretization. We compare SyncTweedies with previous diffusion-synchronization-based methods [5, 18, 35] and MVDiffusion [56], which is finetuned using 3D scenes in the ScanNet [13] dataset. We generate a total of 500 $360°$ panorama images at $0°$ elevation, with a field of view of $72°$.

**Results.** We report quantitative results of the five diffusion synchronization processes discussed in Section 3.2 in Table 4. Table 4 demonstrates a trend consistent with the 1-to-$n$ projection toy experiment results shown in Section 3.4.3. Specifically, SyncTweedies and Case 5, which synchronize the outputs of Tweedie's formula $\phi^{(t)}(\cdot, \cdot)$, exhibit the best performance. Notably, SyncTweedies demonstrates slightly superior performance across all metrics. On the other hand, MVDiffusion [56], which is finetuned using indoor scenes, fails to adapt to new, unseen domains and shows inferior results. The qualitative results are presented in **Appendix** A due to page limit.

Table 5: **A quantitative comparison in 3D Gaussian splats [26] texturing.** KID is scaled by $10^3$. The best in each row is highlighted by **bold**.

| Metric | Diffusion Synchronization | | | | | Optim.-Based | | Iter. View Updating |
| | Case 1 | Sync-Tweedies **Case 2** | Case 3 | Case 4 | Case 5 | SDS [41] | MVDream-SDS [52] | IN2N [19] |
|---|---|---|---|---|---|---|---|---|
| FID [21] ↓ | 211.65 | **106.47** | 120.52 | 114.53 | 116.73 | 110.29 | 141.77 | 109.65 |
| KID [6] ↓ | 85.11 | **14.62** | 19.15 | 17.11 | 18.35 | 19.71 | 38.69 | 15.73 |
| CLIP-S [42] ↑ | 24.69 | **29.55** | 29.53 | 29.30 | 29.12 | 29.33 | 28.69 | 29.25 |

| Prompt | Input 3DGS [26] | Diffusion Synchronization | | | | | Optim.-Based | | Iter. View Updating |
| | | Case 1 | Sync-Tweedies **Case 2** | Case 3 | Case 4 | Case 5 | SDS [41] | MVDream-SDS [52] | IN2N [19] |
|---|---|---|---|---|---|---|---|---|---|
| ''[S*] an intricate wooden carving of a ship'' ''[S*] purple microphone'' | | | | | | | | | |

Figure 5: **Qualitative results of 3D Gaussian splats [26] texturing.** [S*] is a prefix prompt. We use ''Make it to'' for IN2N [19] and ''A photo of'' for the others. SyncTweedies generates high-fidelity textures, while Case 5 lacks fine details due to the variance reduction issue.

## 5.3 3D Gaussian Splats Texturing

Lastly, to verify the difference between SyncTweedies and Case 5 both of which demonstrate applicability up to 1-to-$n$ projections as outlined in Section 3.4.3, we explore texturing 3D Gaussian Splats [26], exemplifying an $n$-to-1 projection case. In 3D Gaussian splats texturing, the projection operation $f_i$ is an $n$-to-1 case, characterized by a volumetric rendering function [25]. This function computes a weighted sum of $n$ 3D Gaussian splats in the canonical space to render a pixel in the instance space. Note that in 3D Gaussian splats texturing, the unprojection $g_i$ and the aggregation $\mathcal{A}$ operation are performed using optimization.

While recent 3D generative models [55, 54] generate plausible 3D objects represented as 3D Gaussian splats, they often lack fine details in the appearance. We validate the effectiveness of SyncTweedies on pretrained 3D Gaussian splats [26] from the Synthetic NeRF dataset [39]. We use 50 views for texture generation and evaluate the results from 150 unseen views. For baselines, we evaluate diffusion-synchronization-based methods, the optimization-based methods, SDS [41], MVDream-SDS [52], and the iterative-view-updating-based method, Instruct-NeRF2NeRF (IN2N) [19].

**Results.** Table 5 and Figure 5 present quantitative and qualitative comparisons of 3D Gaussian splats [26] texturing. SyncTweedies, unaffected by the variance decrease issue, outperforms Case 5 both quantitatively and qualitatively, which is consistent with the observations from the toy experiments in Section 3.4.4. When compared to other baselines based on optimization (SDS [41] and MVDream-SDS [52]) and iterative view updating (IN2N [19]), ours outperforms across all metrics, especially by a large margin in FID [21]. As shown in Figure 5, optimization-based methods tend to generate textures with high saturation, while the iterative-view-updating-based method produces textures lacking fine details. Additional qualitative results are shown in **Appendix** A.

## 6 Conclusion

We have explored various scenarios of diffusion synchronization and evaluated their performance across a range of applications, including ambiguous image generation, panorama generation, and texturing on 3D mesh and 3D Gaussian splats. Our analysis shows that SyncTweedies, which averages the outputs of Tweedie's formula while conducting denoising in multiple instance spaces, offers the best performance and the widest applicability.

**Limitations and Societal Impacts.** Despite the superior performance of SyncTweedies across diverse applications, updating both the geometry and appearance of 3D objects remains an open problem. Also, since the pretrained image diffusion model may have been trained with uncurated images, SyncTweedies might inadvertently produce harmful content.

# Acknowledgments

Thank you to Phillip Y. Lee for valuable discussions on diffusion synchronization, and to Jisung Hwang for providing the 3D mesh renderer. This work was supported by the NRF grant (RS-2023-00209723), IITP grants (RS-2022-II220594, RS-2023-00227592, RS-2024-00399817), and KEIT grant (RS-2024-00423625), all funded by the Korean government (MSIT and MOTIE), as well as grants from the DRB-KAIST SketchTheFuture Research Center, NAVER-Intel Co-Lab, Hyundai NGV, KT, and Samsung Electronics.

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

# Appendix

**Table of Contents**

# A   Qualitative Results

## A.1   3D Mesh Texturing

As shown in Figure 6, SyncTweedies and SyncMVD [35] generate the most realistic output images, aligning with the results of $n$-to-1 projection scenarios discussed in Section 3.4.3. Notably, Paint3D [63], a finetuning-based method, produces inferior textures, losing fine-details, as seen in the appearance of the clock in row 3 and the patterns of the ladybug in row 5. This demonstrates the challenge of acquiring a sufficient amount of high-quality texture images for satisfactory results. The optimization-based method [62] tends to produce images with high-contrast, unnatural colors, as evidenced in rows 4 and 6. Lastly, the iterative-view-updating-based methods [44, 10] show inconsistencies across views noticeable in the dumpster in row 1 and the television in row 9.

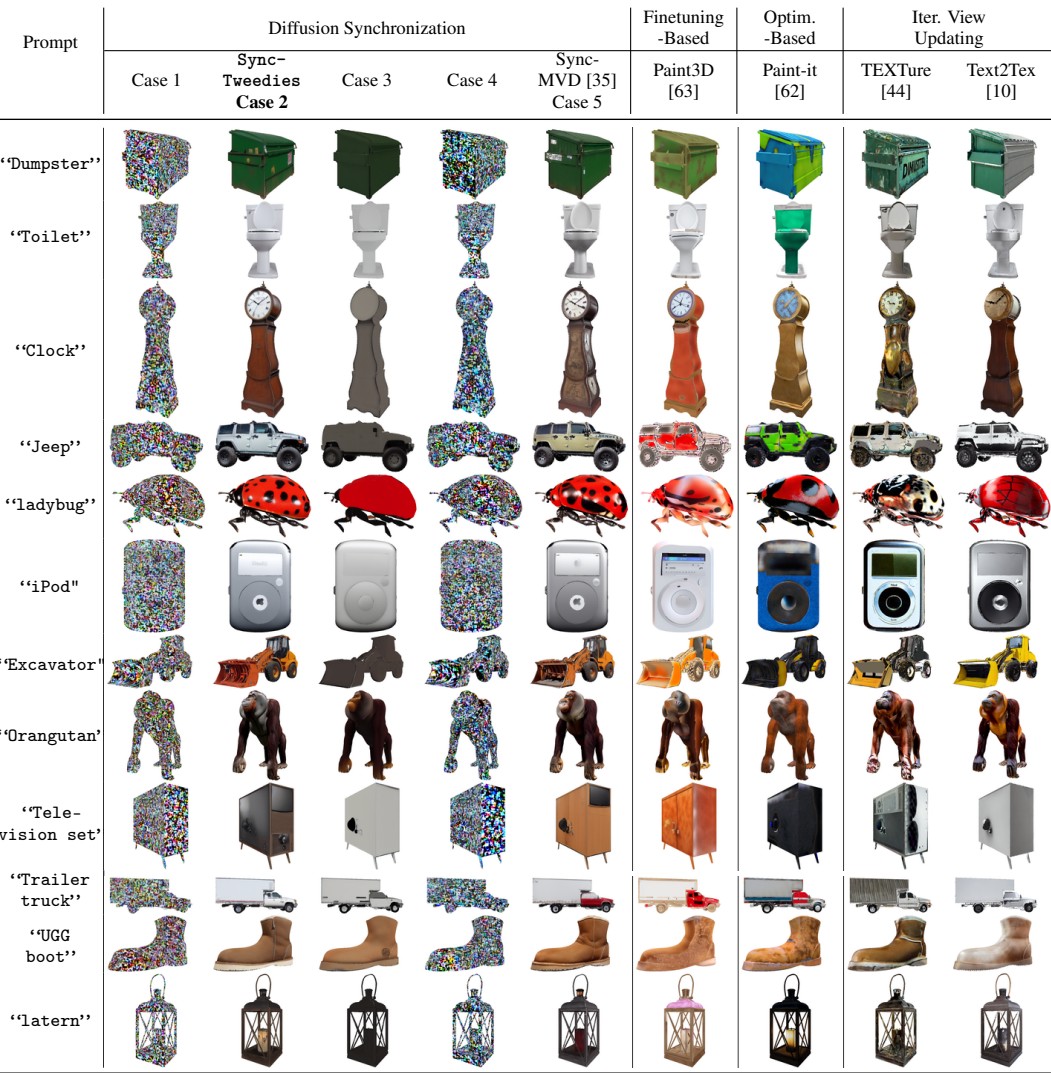

Figure 6: **Qualitative results of 3D mesh texturing.** SyncTweedies and SyncMVD [35] exhibit comparable results, outperforming other baselines. Finetuning-based method [63] produces images without fine details as it was trained on a dataset with coarse texture images. The optimization-based method [62] tends to produce unrealistic and high saturation textures, while iterative-view-updating-based methods [10, 44] show view inconsistencies.

## A.2 Depth-to-360-Panorama Generation

As shown in Figure 7, `SyncTweedies` and Case 5 demonstrate the best results, aligning well with the input depth maps, with `SyncTweedies` showing a slightly better alignment as indicated by the red arrow in Figure 7. On the other hand, MVDiffusion [56], which is finetuned with the depth maps of indoor 3D scenes from the ScanNet [13] dataset, produces suboptimal results and fails to generate realistic $360°$ panoramas for out-of-domain scenes. This demonstrates that MVDiffusion [56] is overfitting to the scenes encountered during finetuning, resulting in a loss of generalizability. Cases 1 and 4, which aggregate the predicted noise $\epsilon_\theta(\cdot)$, produce noisy outputs. Case 3 yields suboptimal panoramas, characterized by monochromatic appearances and a lack of detail.

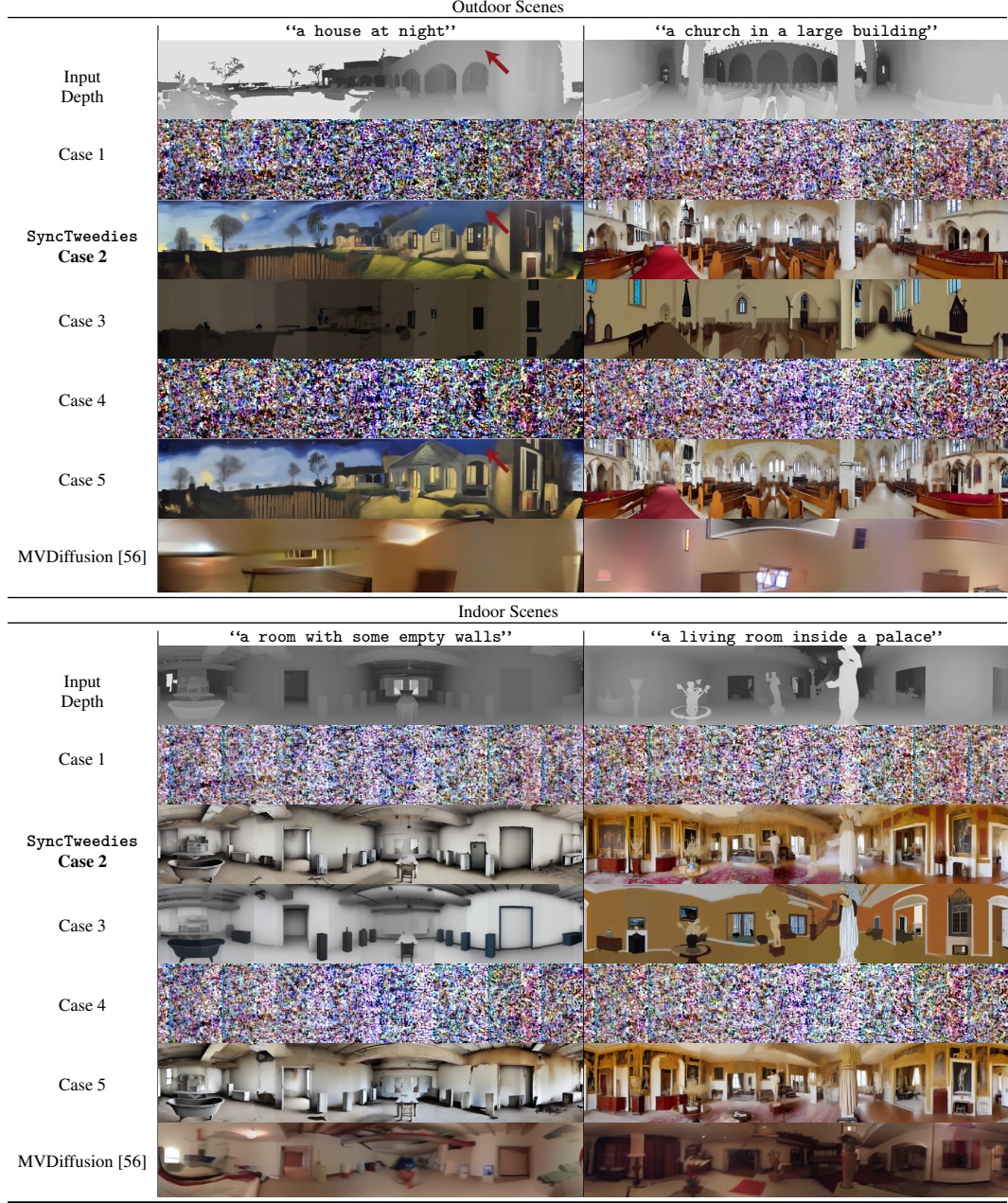

Figure 7: **Qualitative results of depth-to-360-panorama generation.** `SyncTweedies` and Case 5 generate consistent and high-fidelity panoramas as observed in the $1$-to-$n$ projection experiment in Section 3.4.3. MVDiffusion [56] fails to generalize to out-of-domain scenes and generates suboptimal panoramas.

## A.3 3D Gaussian Splats Texturing

Figure 8 shows that `SyncTweedies` generates high-fidelity results with intricate details, such as the carvings of an excavator in row 2, while Case 5 lacks fine details. Optimization-based methods, SDS [41] and MVDream-SDS [52], produce artifacts characterized by high saturation, such as the corns in row 1 and the carrots in row 4. Notably, a finetuning-based method, MVDream-SDS [52], shows inferior quality to SDS. As discussed in Section 3.3.2, the poor quality of textures in the finetuning dataset [16] results in quality degradation. Iterative-view-updating-based method, IN2N [19], fails to preserve fine details, such as the head of the microphone in row 7.

| Prompt | Input 3DGS [26] | Diffusion Synchronization | | | | | Optim.-Based | | Iter. View Updating |
| --- | --- | --- | --- | --- | --- | --- | --- | --- | --- |
| | | Case 1 | Sync-Tweedies **Case 2** | Case 3 | Case 4 | Case 5 | SDS [41] | MVDream-SDS [52] | IN2N [19] |
| ''[S*] corn'' | | | | | | | | | |
| ''[S*] a wooden carving of a excavator'' | | | | | | | | | |
| ''[S*] a drum kit made of ruby'' | | | | | | | | | |
| ''[S*] carrots'' | | | | | | | | | |
| ''[S*] a military ship at sea'' | | | | | | | | | |
| ''[S*] a leather chair'' | | | | | | | | | |
| ''[S*] a wooden carving of a microphone'' | | | | | | | | | |

Figure 8: **Qualitative results of 3D Gaussian splats [26] texturing.** [S*] is a prefix prompt. We use ''`Make it to`'' for IN2N [19] and ''`A photo of`'' for the other methods. Case 5 tends to lose details due to the variance decrease issue, whereas `SyncTweedies` generates realistic images by avoiding this issue. The optimization-based methods [41, 52] produce high contrast, unnatural colors, and the iterative view updating method [19] yields suboptimal outputs due to error accumulation.

# B Details on Experiments

In this section, we provide details on the experiments discussed in Section 5 of the main paper. For all diffusion synchronization processes, we use a fully deterministic DDIM [53] sampling with 30 steps, unless specified otherwise.

We use DeepFloyd [14] as the pretrained diffusion model for the ambiguous image generation which denoises images in the pixel space. For the depth-to-360-panorama generation, 3D mesh texturing, and 3D Gaussian splats texturing, we employ a pretrained depth-conditioned ControlNet [64] which is based on a latent diffusion model, specifically Stable Diffusion [47]. For applications utilizing ControlNet, synchronization during the intermediate steps of diffusion synchronization processes occurs within the same latent space, except for 3D Gaussian splats texturing. In the case of 3D Gaussian splats texturing, synchronization takes place in the RGB space, and detailed explanations are provided in Section B.4.

In the 1-to-$n$ projection cases, each instance space sample is unprojected into the canonical space, resulting in $N$ unprojected samples, $\{g_i(\mathbf{w}_i^{(t)})\}_{i=1}^N$, where $N$ is the number of views. The canonical space sample $\mathbf{z}^{(t)}$ is then obtained by averaging these unprojected samples. The averaging can be weighted based on the visibility from each view. An illustration of the process is shown in Figure 9.

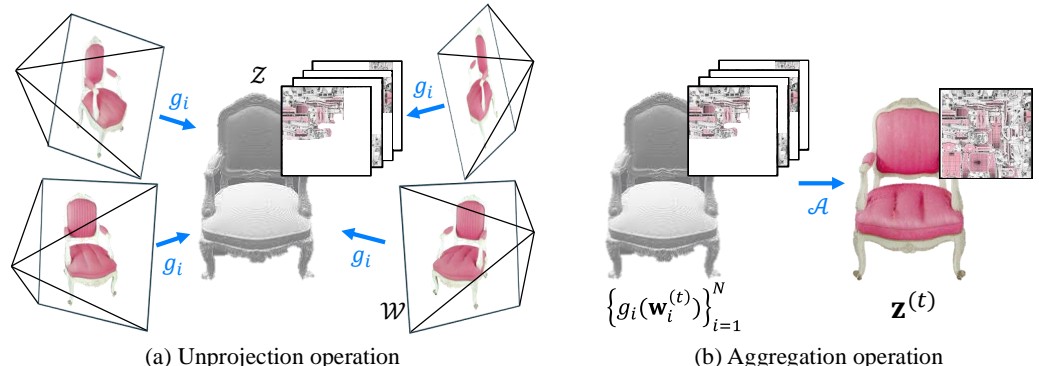

(a) Unprojection operation        (b) Aggregation operation

Figure 9: **Illustration of unprojection and aggregation operation.** The figure shows the synchronization process using the 3D mesh texturing application as an example. The left figure depicts the unprojection operation, where the instance space variables are unprojected into the canonical space. The right figure illustrates the aggregation operation, where the unprojected samples are averaged in the canonical space.

**Evaluation Metrics.** For all applications, we evaluate diversity and fidelity of the generated images using FID [21] and KID [6]. These metrics compute scores based on the distance between the distribution of the generated image set and that of the reference image set, with the reference set forming the target distribution. Refer to each application section for detailed description of constructing the generated image set and the reference image set.

To evaluate the text alignment of the generated images, we report CLIP similarity score [42] (CLIP-S) which measures the similarity between the generated images $\mathbf{w}_i^{(0)}$ and their corresponding text prompts $p_i$ in CLIP [42] embedding space. Additionally, in the ambiguous image generation, we report CLIP alignment score (CLIP-A) and CLIP concealment score (CLIP-C) following previous work, Visual Anagrams [18]. To compute the metrics, we begin by calculating a CLIP similarity matrix $\mathbf{S} \in \mathbb{R}^{N \times N}$ from $N$ pairs of transformations and text prompts:

$$\mathbf{S}_{ij} = E_{\text{img}}(f_i(\mathbf{z}^{(0)}))^T E_{\text{text}}(p_j), \tag{7}$$

where $E_{\text{img}}(\cdot)$ and $E_{\text{text}}(\cdot)$ are the image encoder and the text encoder of the pretrained CLIP model [42], respectively. CLIP-A quantifies the worst alignment among the corresponding image-text pairs, specifically computed as $\min \text{diag}(\mathbf{S})$. However, this metric does not account for misalignment failure cases, where $p_i$ is visualized in $\mathbf{w}_j^{(0)}$ for $i \neq j$. CLIP-C considers alignment of an (a) image (prompt) to all prompts (images) by normalizing the similarity matrix $\mathbf{S}$ using softmax:

$$\frac{1}{N}\text{tr}(\text{softmax}(\mathbf{S}/\tau)), \tag{8}$$

where $\text{tr}(\cdot)$ denotes the trace of a matrix, and $\tau$ is the temperature parameter of CLIP [42]. We set $\tau$ to 0.07.

## B.1 Details on Ambiguous Image Generation — Section 3.4.1

We present the details of the ambiguous image generation experiments in Section 3.4.1. Quantitative and qualitative results are presented in Table 1 and Figure 3.

**Evaluation Setup.** To evaluate the fidelity of the generated images using FID [21] and KID [6], we create a reference set consisting of 5,000 generated images from Stable Diffusion 1.5 [47] with the same text prompts used in the generation of ambiguous images.

**Implementation Details.** We use DeepFloyd [14] which is a two-stage cascaded pixel-space diffusion model. In the first stage, we generate $64 \times 64$ images that are upscaled to $256 \times 256$ images in the subsequent stage.

**Definition of Operations.** In the context of ambiguous image generation, both the instance variables $\{\mathbf{w}_i\}_{i=1:N}$ and canonical variables $\mathbf{z}$ share the same image space. However, instance variables exhibit different appearances from the canonical variable upon applying certain transformations.

In the 1-to-1 projection case, we use the 10 transformations used in Visual Anagrams [18], all of which are 1-to-1 mappings. The projection operation $f_i$ is defined as the transformation itself, and the unprojection operation $g_i$ is defined as the inverse of the transformation matrix.

In the scenario of 1-to-$n$ projection, we employ inner circle rotation as the projection operation $f_i$. This involves rotating the pixels within an inner circle of an image while keeping the outer pixels unchanged. The unprojection operation $g_i$ is the inverse of $f_i$. We use 14 inner circle rotation transformations, with rotation angles evenly spaced in the range $[45°, 175°]$. For evaluation, we utilize the same 95 prompts as in the 1-to-1 case for each transformation, generating $14 \times 95 = 1,350$ ambiguous images. After applying a rotation transformation, the grid of the rotated image does not align with the original image grid. Thus, we use the nearest-neighbor sampling to retrieve pixel colors from the original image to the rotated image. This sampling process leads to a scenario where a single pixel in the original image $\mathbf{z}$ can be mapped to multiple pixels in the rotated image $\mathbf{w}_i$, which is an 1-to-$n$ mapping.

For $n$-to-1 projection, we use the same transformations and text prompts as in the 1-to-1 projection experiment, thus resulting in a total of $10 \times 95 = 950$ ambiguous images. The only difference from the 1-to-1 projection experiment is that the canonical space variable $\mathbf{z}$ is now represented as multiplane images (MPI) [57], where a collection of planes $\{\mathbf{p}_j\}_{j=1:M}$ represents a single canonical variable. Specifically, we compute $\mathbf{z}$ by averaging the multiplane images: $\mathbf{z} = \frac{1}{M} \sum_{j=1}^{M} \mathbf{p}_j$. In the context of $n$-to-1 projection, we substitute the sequence of the unprojection $g_i$ and the aggregation $\mathcal{A}$ operation with an optimization process. The multiplane images $\{\mathbf{p}_j\}$ are optimized using the following objective function:

$$\min_{\{\mathbf{p}_j\}} \sum_{i}^{N} \left| f_i \left( \frac{1}{M} \sum_{j=1}^{M} \mathbf{p}_j \right) - \mathbf{w}_i \right|, \tag{9}$$

where wet set the number of planes $M = 10$.

## B.2 Details on 3D Mesh Texturing — Section 5.1

We provide details of the 3D mesh texturing experiments presented in Section 5.1. Quantitative and qualitative results are shown in Table 3 and Figure 6.

**Evaluation Setup.** We use 429 mesh and prompt pairs collected from previous works, TEXTure [44] and Text2Tex [10]. For texture generation, we use eight views sampled around the object with $45°$ intervals at $0°$ elevation. Two additional views are sampled at $0°$ and $180°$ azimuths with $30°$ elevation. For evaluation, we render each 3D mesh to ten perspective views with randomly sampled azimuths at $0°$ elevation, resulting $10 \times 429 = 4,290$ images. Following SyncMVD [35], the reference set images are generated by ControlNet [64] using the same depth maps and text prompts used in the texture generation.

**Implementation Details.** The resolution of the latent texture image is $1,536 \times 1,536$, and that of the latent perspective view images is $96 \times 96$. In the RGB space, the resolution of the texture image is $1,024 \times 1,024$ and that of the perspective view images is $768 \times 768$.

We adopt two approaches introduced in SyncMVD [35]: Voronoi-diagram-based filling [2] and modified self-attention layers. First, the high resolution of the latent texture image results in a texture image with sparse pixel distribution. To address this issue, we propagate the unprojected pixels to the visible regions of the texture image using the Voronoi-diagram-based filling. Second, spatially distant views tend to generate inconsistent outputs. Therefore, we adopt the modified self-attention mechanism that attends to other views when computing the attention output.

**Definition of Operations.** In the 3D mesh texturing, the canonical variable $\mathbf{z}$ is the texture image of a 3D mesh, and the instance variables $\{\mathbf{w}_i\}_{i=1:N}$ are rendered images from the 3D mesh. The projection operation $f_i$ is a rendering function where nearest-neighbor sampling is utilized to retrieve the color from the texture image to perspective view images.

The unprojection operation $g_i$ is performed using optimization where the texture image $\mathbf{z}$ is updated to minimize the rendering loss with the multi-view images $\{\mathbf{w}_i\}_{i=1:N}$. The projection operation of 3D mesh texturing may involve mapping one pixel in the texture image $\mathbf{z}$ to multiple pixels in a rendered image $\mathbf{w}_i$. Hence, this application corresponds to the 1-to-$n$ projection case as in Section 3.4.3.

### B.3 Details on Depth-to-360-Panorama Generation — Section 5.2

We provide details of the depth-to-360-panorama generation experiments presented in Section 5.2. Refer to Table 4 and Figure 7 for quantitative and qualitative results.

**Evaluation Setup.** We evaluate `SyncTweedies` and the baselines on 500 pairs of $360°$ panorama images and depth maps randomly sampled from the 360MonoDepth [43] dataset. For each $360°$ panorama image, we generate a text prompt using the output of BLIP [30] by providing a perspective view image of the panorama as input.

In the $360°$ panorama generation, we use eight perspective views by evenly sampling azimuths with $45°$ intervals at $0°$ elevation. Each perspective view has a field of view of $72°$ for diffusion-synchronization-based methods and $90°$ for MVDiffusion [56]. For evaluation, we project the generated $360°$ panorama image to ten perspective views with randomly sampled azimuths at $0°$ elevation and a field of view of $60°$. Similarly, the reference set images are obtained by projecting each ground truth $360°$ panorama image into ten perspective views with azimuths randomly sampled and at $0°$ elevation. In total, we use $500 \times 10 = 5,000$ perspective view images for evaluation.

**Implementation Details.** We set the resolution of a latent panorama image to $2,048 \times 4,096$ and that of the latent perspective view images to $64 \times 64$. In the RGB space, a panorama image has a resolution of $1,024 \times 2,048$, and perspective view images have a resolution of $512 \times 512$. As done in the 3D mesh texturing, we apply the Voronoi-diagram-based filling [2] after each unprojection operation and employ the modified self-attention mechanism.

**Definition of Operations.** In the $360°$ panorama generation, the canonical variable $\mathbf{z}$ represents a $360°$ panorama image, while the instance variables $\{\mathbf{w}_i\}_{i=1:N}$ correspond to perspective views of the panorama. The mappings between the panorama image and the perspective views are computed as follows: First, we unproject the pixels of the perspective view image to the 3D space. Then, we apply two rotation matrices based on the azimuth and elevation angles. The pixels are then reprojected onto the surface of a unit sphere, represented as longitudes and latitudes. These spherical coordinates are finally converted to 2D coordinates on the panorama image.

Given the mappings, the projection operation $f_i$ samples colors from the panorama image using the nearest-neighbor method. Since a single pixel of a panorama image $\mathbf{z}$ can be mapped to multiple pixels of a perspective view image $\mathbf{w}_i$, the $360°$ panorama generation is a 1-to-$n$ projection case, as discussed in Section 3.4.3.

### B.4 Details on 3D Gaussian Splats Texturing — Section 5.3

We provide details of the 3D Gaussian splats texturing experiment presented in Section 5.3. Quantitative and qualitative results are provided in Table 5 and Figure 8.

**Evaluation Setup.** For evaluation, we use 3D Gaussian splats trained with multi-view images from the Synthetic NeRF dataset [39], consisting of 8 objects. We generate 40 textured 3D Gaussian splats by utilizing five different prompts per scene. We use 50 views for texture generation and 150 unseen views for evaluation.

**Implementation Details.** As described in Section B, we employ ControlNet [64] which denoises latent images. To render the latent images, we replace the spherical harmonics coefficients of a 3D Gaussian splats to a 4-channel latent vector. For the optimization, we run 2,000 iterations with a learning rate of 0.025. When applicable, we perform the optimization in RGB space by decoding the latent variables for diffusion-synchronization-based methods.

**Definition of Operations.** The canonical variables $\{\mathbf{z}_j\}_{j=1:M}$ are 3D Gaussian splats and the instance space variables $\{\mathbf{w}_i\}_{i=1:N}$ are the rendered images from the 3D Gaussian splats. The

projection operation $f_i$ is a volume rendering function [25, 26] where the colors (latent vectors) of multiple 3D Gaussian splats are composited to render a pixel. This corresponds to the $n$-to-1 projection as discussed in Section 3.4.4. In 3D Gaussian splats texturing, only the colors of 3D Gaussian splats $\mathbf{z} = \{\mathbf{s}_j\}_{j=1:M}$ are optimized from multi-view images $\{\mathbf{w}_i\}_{i=1:N}$, while keeping other parameters, such as positions, fixed, as done in the $n$-to-1 experiment in Section 3.4.4.

Table 6: **A quantitative comparison in arbitrary-sized image generation.** KID is scaled by $10^3$. For each row, we highlight the column whose value is within 95% of the best.

| Metric | Case 1 | SyncTweedies **Case 2** | MultiDiffusion [5] Case 3 | Case 4 | Case 5 |
|---|---|---|---|---|---|
| FID [21] ↓ | 32.83 | 32.82 | 32.83 | 32.82 | 32.83 |
| KID [6] ↓ | 7.79 | 7.79 | 7.79 | 7.79 | 7.80 |
| CLIP-S [42] ↑ | 31.69 | 31.69 | 31.69 | 31.69 | 31.69 |

## C  Arbitray-Sized Image Generation

In addition to the 1-to-1 projection case presented in Section 3.4.2, we present arbitrary-sized image generation. In contrast to depth-to-360-panorama generation, which corresponds to the 1-to-$n$ projection case, arbitrary-sized image generation is a 1-to-1 projection case.

**Evaluation Setup.** We follow the evaluation setup used in SyncDiffusion [29]. Using Stable Diffusion 2.0 [47] as the pretrained diffusion model, we generate 500 arbitrary-sized images of $512 \times 3,072$ resolution per prompt. With six text prompts from SyncDiffusion [29], we generate a total of $500 \times 6 = 3,000$ arbitrary-sized images. For quantitative evaluation, we report FID [21], KID [6], and CLIP-S [42]. Each generated arbitrary-sized image is randomly cropped to partial view images with $512 \times 512$ resolution. For the reference set, we generate $3,000$ images with a resolution of $512 \times 512$ from the pretrained diffusion model using the same text prompts.

**Implementation Details.** The resolution of latent arbitrary-sized image is $64 \times 384$, and the resolution of an instance space sample is $64 \times 64$. We use deterministic DDIM [53] sampling with 50 steps.

**Definition of Operations.** The projection operation $f_i$ corresponds to cropping a partial view of the arbitrary-sized image, which is a 1-to-1 projection. The unprojection operation $g_i$ is the inverse of the $f_i$ which pastes the partial view image onto the canvas of the arbitrary-sized image.

**Results.** We report quantitative results in Table 6 and qualitative results in Figure 10. As mathematically proven in Section D, the quantitative results show that all diffusion synchronization cases exhibit comparable performances, which aligns with the observations from the 1-to-1 experiment in Section 3.4.2. This is further supported by the qualitative results in Figure 10, where all cases produce identical arbitrary-sized images, indicating that any option can be used when the projection is 1-to-1.

**Results using Gaudi Intel-v2.** Additionally, we present qualitative results of arbitrary-sized image generation using Intel Gaudi-v2 in Figure 11, along with a comparison of computation times between Intel Gaudi-v2 and NVIDIA A6000 in Figure 12. We observe that Intel Gaudi-v2 achieves 1.8 to 1.9 times faster runtimes compared to the NVIDIA A6000.

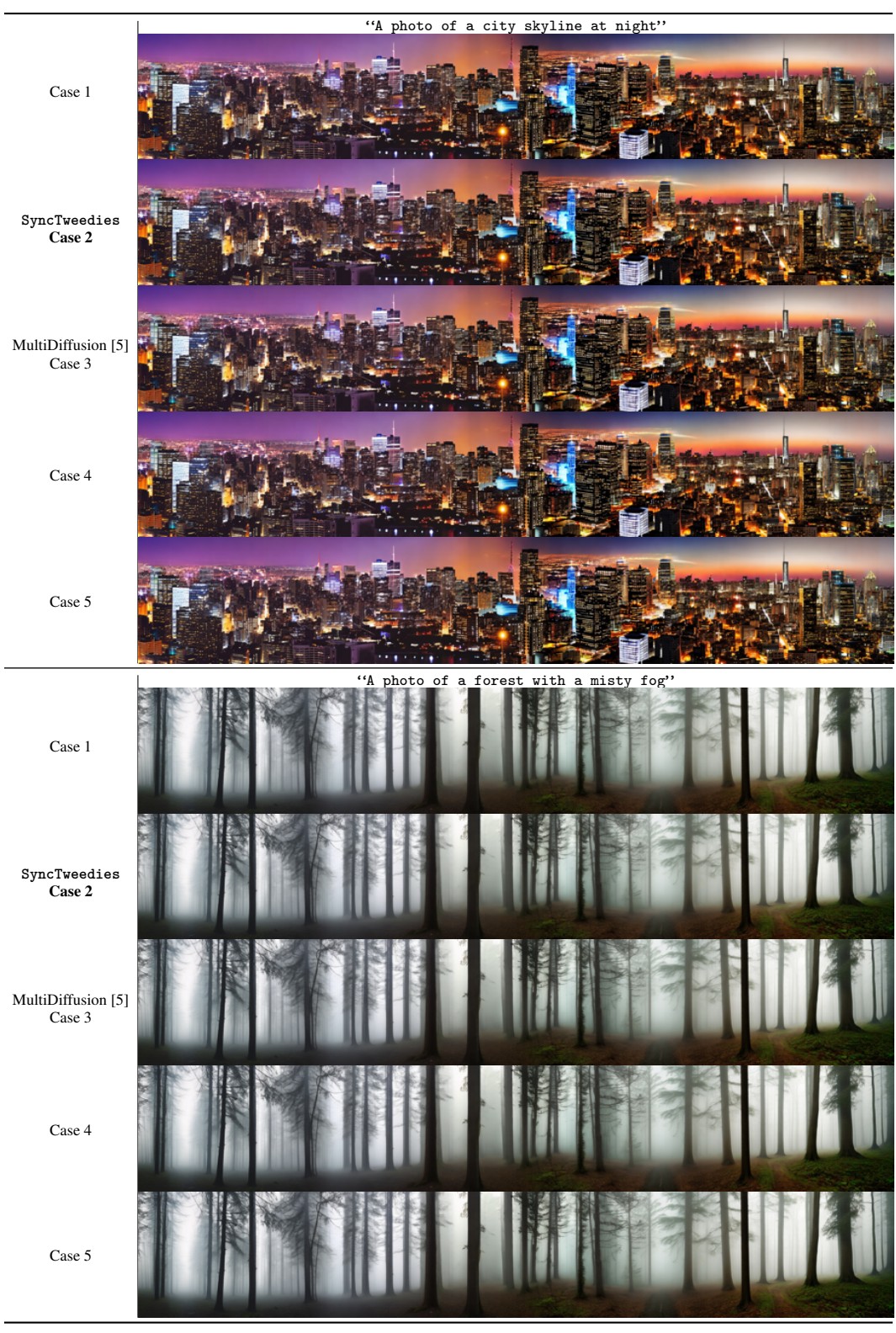

Figure 10: **Qualitative results of arbitrary-sized image generation.** All diffusion synchronization processes generate identical results in the 1-to-1 projection.

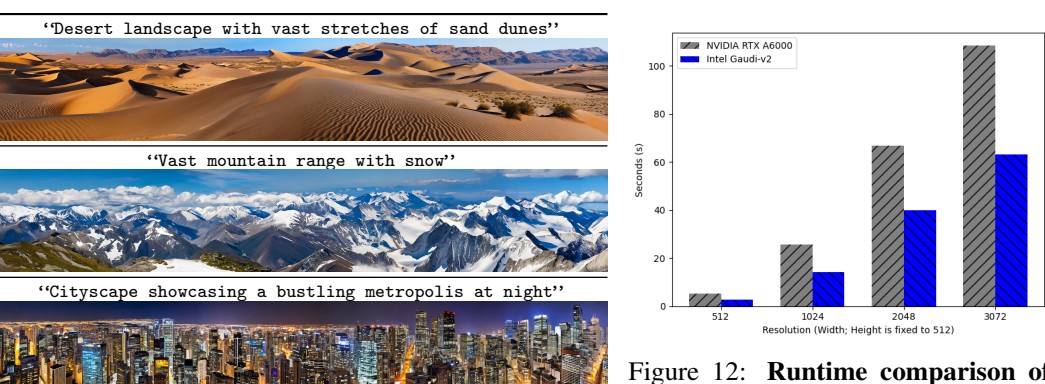

Figure 11: **Qualitative results of arbitrary-sized image generation using Intel Gaudi-v2.** `SyncTweedies` (Case 2) is used for all text prompts.

Figure 12: **Runtime comparison of NVIDIA RTX A6000 and Intel Gaudi-v2.** We use four different width sizes for the arbitrary-sized images: $\{512, 1024, 2048, 3072\}$.

# D  1-to-1 Projection

It is mathematically guaranteed that Cases 1-5 become identical when the mappings are 1-to-1 and noises are initialized by projecting from the canonical space $\mathbf{w}_i^{(T)} = f_i(\mathbf{z}^{(T)})$, where $\mathbf{z}^{(T)} \sim \mathcal{N}(\mathbf{0}, \mathbf{I})$. Note that $\phi^{(t)}(\cdot, \cdot)$ and $\psi^{(t)}(\cdot, \cdot)$ are linear operations and commutative with other linear operation such as $f_i, \mathcal{A}$, and $g_i$. Assume the following conditions hold:

$$\mathbf{z}^{(T)} = \mathcal{A}(\{g_i(f_i(\mathbf{z}^{(T)}))\}), \tag{10}$$

$$\mathcal{A}(\{g_i(\mathbf{w}_i)\}) = \mathcal{A}(\{g_i(f_i(\mathcal{A}(\{g_j(\mathbf{w}_j)\})))\}) \quad \forall\{\mathbf{w}\}_{i=1}^N. \tag{11}$$

Based on induction, we have:

$$\mathbf{z}^{(t-1)} = \psi^{(t)}(\mathbf{z}^{(t)}, \phi^{(t)}(\mathbf{z}^{(t)}, \mathcal{A}(\{g_i(\epsilon_\theta(f_i((\mathbf{z}^{(t)}))))\}))) \quad\quad (\text{Case 4}) \tag{12}$$

$$= \psi^{(t)}(\mathbf{z}^{(t)}, \phi^{(t)}(\mathcal{A}(\{g_i(f_i(\mathbf{z}^{(t)}))\}), \mathcal{A}(\{g_i(\epsilon_\theta(f_i(\mathbf{z}^{(t)})))\}))) \tag{13}$$

$$= \psi^{(t)}(\mathbf{z}^{(t)}, \mathcal{A}(\{g_i(\phi^{(t)}(f_i(\mathbf{z}^{(t)}), \epsilon_\theta(f_i(\mathbf{z}^{(t)}))))\})) \quad\quad (\text{Case 5}) \tag{14}$$

$$= \psi^{(t)}(\mathcal{A}(\{g_i(f_i(\mathbf{z}^{(t)}))\}), \mathcal{A}(\{g_i(\phi^{(t)}(f_i(\mathbf{z}^{(t)}), \epsilon_\theta(f_i(\mathbf{z}^{(t)}))))\})) \tag{15}$$

$$= \mathcal{A}(\{g_i(\psi^{(t)}(f_i(\mathbf{z}^{(t)}), \phi^{(t)}(f_i(\mathbf{z}^{(t)}), \epsilon_\theta(f_i(\mathbf{z}^{(t)})))))\}) \tag{16}$$

$$= \mathcal{A}(\{g_i(f_i(\mathcal{A}(\{g_j(\psi^{(t)}(f_j(\mathbf{z}^{(t)}), \phi^{(t)}(f_j(\mathbf{z}^{(t)}), \epsilon_\theta(f_j(\mathbf{z}^{(t)}))))\})))\}) \tag{17}$$

$$= \mathcal{A}(\{g_i(f_i(\mathbf{z}^{(t-1)}))\}), \tag{18}$$

where the last equality holds the induction hypothesis. This proves that Cases 4-5 are identical.

For instance variable denoising cases we have:

$$\mathbf{w}_i^{(t-1)} = \psi^{(t)}(\mathbf{w}_i^{(t)}, \phi^{(t)}(\mathbf{w}_i^{(t)}, f_i(\mathcal{A}(\{g_j(\epsilon_\theta(\mathbf{w}_j^{(t)}))\})))) \quad\quad (\text{Case 1}) \tag{19}$$

$$= \psi^{(t)}(\mathbf{w}_i^{(t)}, \phi^{(t)}(f_i(\mathcal{A}(\{g_j(\mathbf{w}_j^{(t)})\})), f_i(\mathcal{A}(\{g_j(\epsilon_\theta(\mathbf{w}_j^{(t)}))\})))) \tag{20}$$

$$= \psi^{(t)}(\mathbf{w}_i^{(t)}, f_i(\mathcal{A}(\{g_j(\phi^{(t)}(\mathbf{w}_j^{(t)}, \epsilon_\theta(\mathbf{w}_j^{(t)})))\}))) \quad\quad (\text{Case 2}) \tag{21}$$

$$= \psi^{(t)}(f_i(\mathcal{A}(\{g_j(\mathbf{w}_j^{(t)})\})), f_i(\mathcal{A}(\{g_j(\phi^{(t)}(\mathbf{w}_j^{(t)}, \epsilon_\theta(\mathbf{w}_j^{(t)})))\}))) \tag{22}$$

$$= f_i(\mathcal{A}(\{g_j(\psi^{(t)}(\mathbf{w}_j^{(t)}, \phi^{(t)}(\mathbf{w}_j^{(t)}, \epsilon_\theta(\mathbf{w}_j^{(t)}))))\})) \quad\quad (\text{Case 3}) \tag{23}$$

$$= f_i(\mathcal{A}(\{g_j(f_j(\mathcal{A}(\{g_k(\psi^{(t)}(\mathbf{w}_k^{(t)}, \phi^{(t)}(\mathbf{w}_k^{(t)}, \epsilon_\theta(\mathbf{w}_k^{(t)}))))\})))\})) \tag{24}$$

$$= f_i(\mathcal{A}(\{g_j(\mathbf{w}_j^{(t-1)})\})), \tag{25}$$

where the last equality holds the induction hypothesis. This proves that Cases 1-3 are identical. Lastly, based on the definition of the projection operation, we have:

$$\mathbf{w}_i^{(t-1)} = f_i(\mathbf{z}^{(t-1)}) \tag{26}$$

$$= f_i(\mathcal{A}(\{g_i(\psi^{(t)}(f_i(\mathbf{z}^{(t)}), \phi^{(t)}(f_i(\mathbf{z}^{(t)}), \epsilon_\theta(f_i(\mathbf{z}^{(t)}))))\})) \tag{27}$$

$$= f_i(\mathcal{A}(\{g_j(\psi^{(t)}(\mathbf{w}_j^{(t)}, \phi^{(t)}(\mathbf{w}_j^{(t)}, \epsilon_\theta(\mathbf{w}_j^{(t)}))))\})) \qquad \text{(Case 3)}. \tag{28}$$

This proves that canonical variable denoising cases (Cases 4-5) are equivalent to Case 3.

We validate the proof both qualitatively and quantitatively in applications with 1-to-1 projection: ambiguous image generation and arbitrary-sized image generation in Section 3.4.2 and Section C, respectively, where all cases generate identical results.

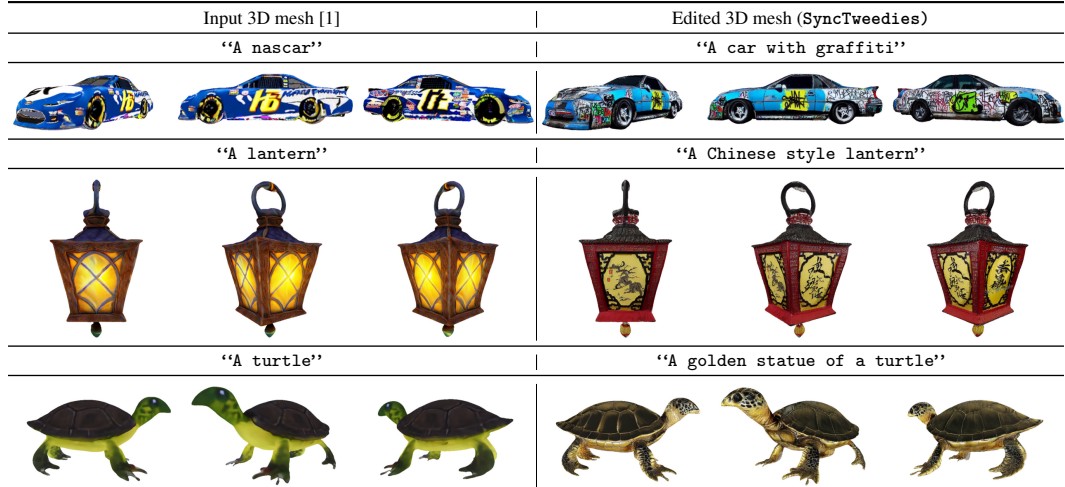

Figure 13: **Qualitative results of 3D mesh texture editing.** We edit the textures of the 3D meshes generated from *Genies* [1] using `SyncTweedies`.

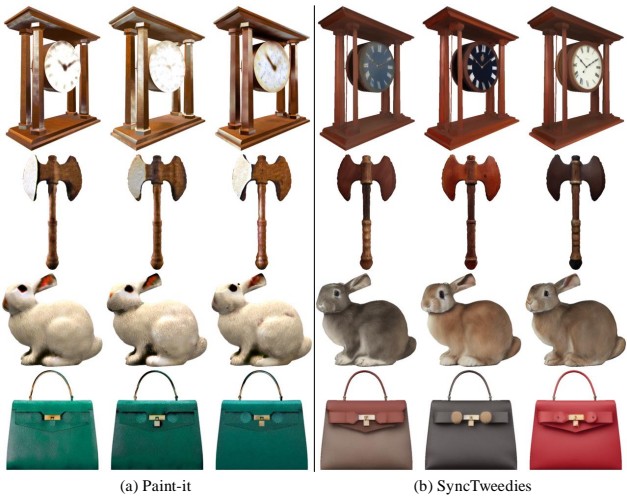

Figure 14: **Diversity comparison.** Optimization-based method Paint-it [62] (Left) and diffusion-synchronization-based method, `SyncTweedies` (Right). `SyncTweedies` generates more diverse images.

# E   3D Mesh Texture Editing and Diversity

In this section, we extend the 3D mesh texture generation from Section 5.1 and present a texture editing application, along with a diversity comparison of `SyncTweedies` to the optimization-based method Paint-it [62].

## E.1   3D Mesh Texture Editing

Despite the recent successes of 3D generation models [1, 34], the textures of the generated 3D meshes often lack fine details. We utilize `SyncTweedies` to edit the textures of the generated 3D meshes, and enhance the texture quality. Specifically, we use the 3D meshes generated from a text-to-3D model, Genie [1].

We follow SDEdit [36] to edit the textures of 3D meshes. We begin by adding noise at an intermediate time $t'$ to the texture image of the 3D mesh and then perform a reverse process starting from $t'$.

**Implementation Details.**   We set the CFG weight [22] to 30 and $t'$ to 0.8. For other settings, we follow the 3D mesh texture generation experiment presented in Section 5.1.

**Results.**   We present qualitative results of 3D mesh texture editing in Figure 13. The 3D meshes edited with `SyncTweedies` exhibit fine details, including graffiti on the car in row 1, paintings on the lantern in row 2, and the intricate shells of the turtle in row 3.

## E.2   Diversity of `SyncTweedies`

In Figure 14, we present qualitative results of 3D mesh texturing using the optimization-based method (Paint-it [62]) and `SyncTweedies` with different random seeds. `SyncTweedies` generates more diverse texture images compared to Paint-it.

# F   Runtime and VRAM Usage Comparison

Table 7: **A runtime comparison in 3D mesh texturing and 3D Gaussian splats texturing applications.** The best in each row is highlighted by **bold**.

| Metric | Diffusion Synchronization | Finetuning. -Based | Optim. -Based | Iter. View Updating | |
|---|---|---|---|---|---|
| | | | 3D Mesh Texturing | | |
| | SyncTweedies **Case 2** | Paint3D [63] | Paint-it [62] | TEXTure [44] | Text2Tex [10] |
| Runtime (minutes) ↓ | 1.83 | 2.65 | 21.95 | **1.54** | 13.10 |
| | | | 3D Gaussian Splats Texturing | | |
| | SyncTweedies **Case 2** | - | SDS [41] | IN2N [19] | |
| | **10.56** | - | 85.50 | 37.93 | |

Table 8: **A VRAM usage comparison in 3D mesh texturing and 3D Gaussian splats texturing applications.** The best in each row is highlighted by **bold**.

| Metric | Diffusion Synchronization | Finetuning -Based | Optim. -Based | Iter. View Updating | |
|---|---|---|---|---|---|
| | | | 3D Mesh Texturing | | |
| | SyncTweedies **Case 2** | Paint3D [63] | Paint-it [62] | TEXTure [44] | Text2Tex [10] |
| VRAM Usage (GiB) ↓ | **6.49** | 9.15 | 28.36 | 10.66 | 10.92 |
| | | | 3D Gaussian Splats Texturing | | |
| | SyncTweedies **Case 2** | - | SDS [41] | IN2N [19] | |
| | 9.30 | - | **6.87** | 10.38 | |

As discussed in Section 4, one of the advantages of diffusion synchronization processes is the fast computational speed. We compare the runtime performance of `SyncTweedies` with optimization-based and iterative-view-updating-based methods in the 3D mesh texturing and the 3D Gaussian splats texturing. The quantitative results are presented in Table 7.

In the 3D mesh texturing application, `SyncTweedies` shows faster running time than other baselines except TEXTure [44] which shows comparable running time. However, TEXTure [44] generates suboptimal texture outputs as observed in Table 3 and Figure 6. The finetuning-based method Paint3D [63] shows a comparable running time to `SyncTweedies`, but it shows inferior quality, as seen in Table 3 and Figure 6. Another iterative-view-updating-based method, Text2Tex [10], improves quality of texture image by integrating a refinement module, but this comes at the cost of additional overhead in terms of running time. In contrast, `SyncTweedies` achieves running times that are 7 times faster than Text2Tex and even outperforms across all metrics as shown in Table 3. Lastly, `SyncTweedies` shows 11 times faster running time when compared to Paint-it [62], an optimization-based method.

In the 3D Gaussian splats texturing, `SyncTweedies` achieves the fastest running time. `SyncTweedies` is 3 times faster than the iterative-view-updating-based method IN2N [19], and 8 times faster than the optimization-based method, SDS [41]. This shows that `SyncTweedies` not only generates high-fidelity textures, but also excels other baselines in computational speed. We use the NVIDIA RTX A6000 for the runtime comparisons.

Additionally, in Table 8, we present results comparing the VRAM usage of `SyncTweedies` and the baselines, where our `SyncTweedies` requires around 6-9 GiB of memory, making it suitable for most GPUs.

# G   User Study

We conduct user studies to evaluate the textures of the generated 3D Gaussian splats [26] through Amazon's Mechanical Turk. Following the methodology of Ritchie [45], participants were presented with input text prompts and randomly sampled output images generated by our method and the baseline methods. Participants are asked to choose the most plausible image that aligns with the given text prompt. In Table 9, our results are the most preferred in the human evaluations compared to the other baselines.

**Details on User Study.**   We conduct separate user studies comparing our method to diffusion-synchronization-based methods, optimization-based methods (SDS [41], MVDream-SDS [52]), and iterative-view-updating-based method (IN2N [19]). For each user study, we use 20 images in a shuffled order including five vigilance tasks. We collect survey responses only from participants who pass the vigilance tasks. Specifically, 94 out of 100 participants passed in the test with Case 5, 90 out of 100 passed with SDS [41], 95 out of 100 passed with MVDream-SDS [52], and 92 out of 100 passed with IN2N [19]. Screenshots of the user study, including an example of vigilance tasks, are shown in Figure 15.

Table 9: **User study results in 3D Gaussian splats texturing application.** `SyncTweedies` is the most preferred method over the baselines from human evaluators.

| Baselines | Case 5 | SDS [41] | MVDream-SDS [52] | IN2N [19] |
|---|---|---|---|---|
| Prefer Baseline (%) | 33.56 | 41.33 | 12.21 | 40.05 |
| Prefer `SyncTweedies` (%) | **66.44** | **58.67** | **87.79** | **59.95** |

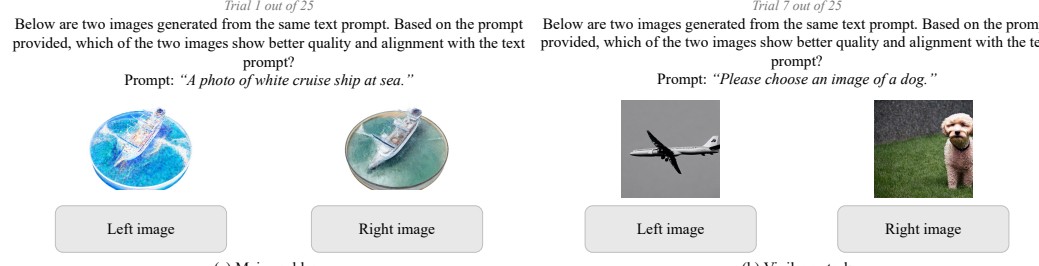

(a) Main problem            (b) Vigilance task

Figure 15: **3D Gaussian splats texturing user study screenshots.** A screenshot of a main problem (left) and a vigilance task (right) is shown.

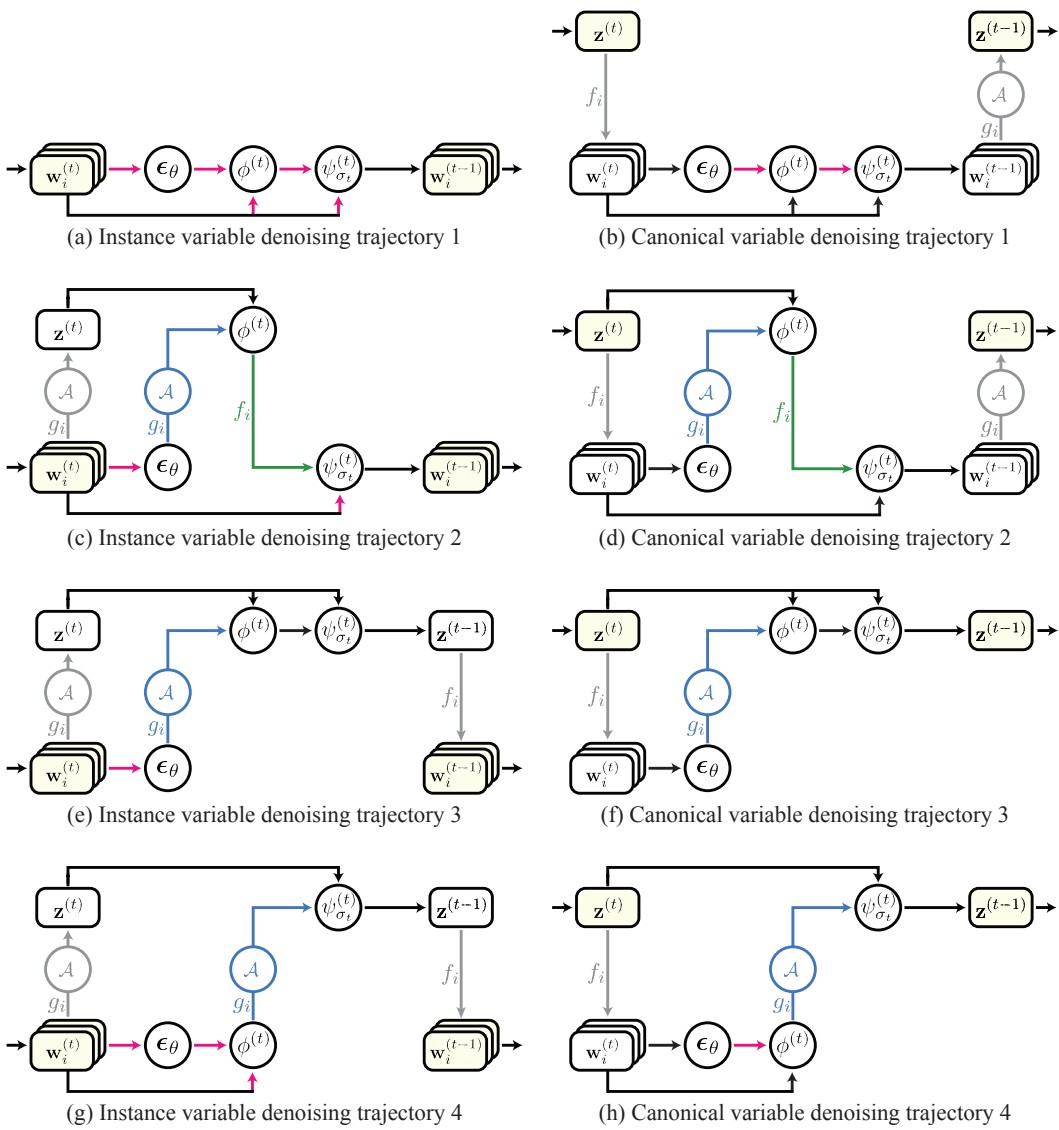

(a) Instance variable denoising trajectory 1      (b) Canonical variable denoising trajectory 1

(c) Instance variable denoising trajectory 2      (d) Canonical variable denoising trajectory 2

(e) Instance variable denoising trajectory 3      (f) Canonical variable denoising trajectory 3

(g) Instance variable denoising trajectory 4      (h) Canonical variable denoising trajectory 4

Figure 16: **Diagrams of diffusion synchronization processes.** All feasible trajectories of the instance variable denoising process (left) and the canonical variable denoising process (right). Each row shares the same trajectory with different variables denoised.

# H   Analysis of Diffusion Synchronization Processes

As outlined in Section 3.4.4, we present a comprehensive analysis of all possible diffusion synchronization processes, including the representative five diffusion synchronization processes introduced in Section 3.2. Following the main paper, we categorize diffusion synchronization processes into two types: the *instance variable denoising process*, where instance variables $\{\mathbf{w}_i^{(t)}\}$ are denoised, and the *canonical variable denoising process*, which denoises a canonical variable $\mathbf{z}^{(t)}$ directly. Unlike the representative cases, other all feasible cases either take inconsistent inputs when computing $\boldsymbol{\epsilon}_\theta(\cdot)$, $\phi^{(t)}(\cdot,\cdot)$ and $\psi^{(t)}(\cdot,\cdot)$ or conduct the aggregation $\mathcal{A}$ multiple times. Additionally, for a more exhaustive analysis, we introduce another type of diffusion synchronization processes, named the *combined variable denoising process*, which denoises $\{\mathbf{w}_i^{(t)}\}$ and $\mathbf{z}^{(t)}$ together.

We present a total of 46 feasible cases for the instance variable denoising process, 8 for the canonical variable denoising process, and an additional 6 representative cases for the combined variable denoising process. We provide instance variable denoising cases in Section H.2, and canonical variable denoising cases in Section H.3. Additionally, the six representative cases for the combined variable denoising process are detailed in Section H.4. We conduct a quantitative comparison of all listed cases following the experiment setup outlined in Section 3.4.1, and the results are presented in Section H.5.

## H.1   Overview

We provide the representative trajectories in Figure 16, where (a)-(b), (c)-(d), (e)-(f), and (g)-(h) follow the same trajectory but differ in the denoising variable, either instance or canonical, respectively. In each denoising case, there are $2^2 = 4$ possible trajectories determined by whether $\phi^{(t)}(\cdot,\cdot)$ and $\psi^{(t)}(\cdot,\cdot)$ are computed in the canonical space or instance space. This is because among the three computation layers—$\boldsymbol{\epsilon}_\theta(\cdot)$, $\phi^{(t)}(\cdot,\cdot)$ and $\psi^{(t)}(\cdot,\cdot)$—only the last two operations can be computed in both the canonical space and the instance space unlike noise prediction which is only available in the instance space. Table 10 summarizes the computation spaces of $\phi^{(t)}(\cdot,\cdot)$ and $\psi^{(t)}(\cdot,\cdot)$, along with their corresponding trajectories.

Table 10: **Computation space of each denoising trajectory.** $\phi^{(t)}(\cdot,\cdot)$ and $\psi^{(t)}(\cdot,\cdot)$ can be computed in both instance space $\mathcal{W}_i$ or canonical space $\mathcal{Z}$, whereas noise prediction $\boldsymbol{\epsilon}_\theta(\cdot)$ can only be computed in the instance space.

| Trajectory | $\phi^{(t)}(\cdot,\cdot)$
Computation space | $\psi^{(t)}(\cdot,\cdot)$
Computation space |
|---|---|---|
| Trajectory 1 | $\mathcal{W}_i$ | $\mathcal{W}_i$ |
| Trajectory 2 | $\mathcal{Z}$ | $\mathcal{W}_i$ |
| Trajectory 3 | $\mathcal{Z}$ | $\mathcal{Z}$ |
| Trajectory 4 | $\mathcal{W}_i$ | $\mathcal{Z}$ |

Next, we introduce an additional operator $\mathcal{F}_i$ that synchronizes instance variables. This operator unprojects a set of instance variables and averages them in the canonical space. Subsequently, the aggregated variables are reprojected to the instance space:

$$\mathcal{F}_i(\{\mathbf{w}_j\}_{j=1:N}) = f_i(\mathcal{A}(\{g_j(\mathbf{w}_j)\}_{j=1:N})). \tag{29}$$

The red arrows in the diagrams of Figure 16 indicate the potential incorporation of $\mathcal{F}_i$. Thus, a total of $2^N$ different cases can be derived from a trajectory marked by $N$ red arrows, depending on whether $\mathcal{F}_i$ is applied to each variable or not.

Lastly, we review the five representative diffusion synchronization processes discussed in Section 3.2, along with two additional denoising processes: an instance variable denoising process that proceeds without synchronization (No Synchronization) and a canonical variable denoising process that averages the outputs of $\psi^{(t)}(\cdot,\cdot)$ (Case 6):

$$\text{No Synchronization}\ : \mathbf{w}_i^{(t-1)} = \psi^{(t)}(\mathbf{w}_i^{(t)}, \phi^{(t)}(\mathbf{w}_i^{(t)}, \boldsymbol{\epsilon}_\theta(\mathbf{w}_i^{(t)})))$$

$$\text{Case 1}\ : \mathbf{w}_i^{(t-1)} = \psi^{(t)}(\mathbf{w}_i^{(t)}, \phi^{(t)}(\mathbf{w}_i^{(t)}, \mathcal{F}_i(\boldsymbol{\epsilon}_\theta(\mathbf{w}_i^{(t)}))))$$

$$\text{Case 2}\ : \mathbf{w}_i^{(t-1)} = \psi^{(t)}(\mathbf{w}_i^{(t)}, \mathcal{F}_i(\phi^{(t)}(\mathbf{w}_i^{(t)}, \boldsymbol{\epsilon}_\theta(\mathbf{w}_i^{(t)}))))$$

$$\text{Case 3} : \mathbf{w}_i^{(t-1)} = \mathcal{F}_i(\psi^{(t)}(\mathbf{w}_i^{(t)}, \phi^{(t)}(\mathbf{w}_i^{(t)}, \epsilon_\theta(\mathbf{w}_i^{(t)}))))$$

$$\text{Case 4} : \mathbf{z}^{(t-1)} = \psi^{(t)}(\mathbf{z}^{(t)}, \phi^{(t)}(\mathbf{z}^{(t)}, \mathcal{A}(\{g_i(\epsilon_\theta(f_i(\mathbf{z}^{(t)})))\})))$$

$$\text{Case 5} : \mathbf{z}^{(t-1)} = \psi^{(t)}(\mathbf{z}^{(t)}, \mathcal{A}(\{g_i(\phi^{(t)}(f_i(\mathbf{z}^{(t)}), \epsilon_\theta(f_i(\mathbf{z}^{(t)}))))\}))$$

$$\text{Case 6} : \mathbf{z}^{(t-1)} = \mathcal{A}(\{g_i(\psi^{(t)}(f_i(\mathbf{z}^{(t)}), \phi^{(t)}(f_i(\mathbf{z}^{(t)}), \epsilon_\theta(f_i(\mathbf{z}^{(t)})))))\}).$$

Note that Case 3 and 6 are identical except for the initialization, which can be either $\{\mathbf{w}_i^{(T)}\}$ or $\mathbf{z}^{(T)}$. For the independent instance variable denoising process (No Synchronization), synchronization is only applied at the end of the denoising process.

## H.2 Instance Variable Denoising Process

Here, we explore all possible instance variable denoising processes. Here, the canonical space $\mathcal{Z}$ is employed to *synchronize* the outputs of $\epsilon_\theta(\cdot)$, $\phi^{(t)}(\cdot, \cdot)$ and $\psi^{(t)}(\cdot, \cdot)$ in the instance spaces.

Following the trajectory 1 shown in part (a) of Figure 16, marked by five red arrows, there are a total of $2^5 = 32$ possible denoising processes. This includes the independent instance variable denoising process (No Synchronization), where $\mathcal{F}_i$ is not applied at any red arrow. Additionally, the three representative instance variable denoising processes, Cases 1-3, are also included, along with Cases 7-34 which are presented below:

$$\text{Case 7} : \mathbf{w}_i^{(t-1)} = \psi^{(t)}(\mathbf{w}_i^{(t)}, \phi^{(t)}(\mathbf{w}_i^{(t)}, \epsilon_\theta(\mathcal{F}_i(\mathbf{w}_i^{(t)}))))$$

$$\text{Case 8} : \mathbf{w}_i^{(t-1)} = \psi^{(t)}(\mathbf{w}_i^{(t)}, \phi^{(t)}(\mathbf{w}_i^{(t)}, \mathcal{F}_i(\epsilon_\theta(\mathcal{F}_i(\mathbf{w}_i^{(t)})))))$$

$$\text{Case 9} : \mathbf{w}_i^{(t-1)} = \psi^{(t)}(\mathbf{w}_i^{(t)}, \phi^{(t)}(\mathcal{F}_i(\mathbf{w}_i^{(t)}), \epsilon_\theta(\mathbf{w}_i^{(t)})))$$

$$\text{Case 10} : \mathbf{w}_i^{(t-1)} = \psi^{(t)}(\mathbf{w}_i^{(t)}, \phi^{(t)}(\mathcal{F}_i(\mathbf{w}_i^{(t)}), \epsilon_\theta(\mathcal{F}_i(\mathbf{w}_i^{(t)}))))$$

$$\text{Case 11} : \mathbf{w}_i^{(t-1)} = \psi^{(t)}(\mathbf{w}_i^{(t)}, \phi^{(t)}(\mathcal{F}_i(\mathbf{w}_i^{(t)}), \mathcal{F}_i(\epsilon_\theta(\mathbf{w}_i^{(t)}))))$$

$$\text{Case 12} : \mathbf{w}_i^{(t-1)} = \psi^{(t)}(\mathbf{w}_i^{(t)}, \phi^{(t)}(\mathcal{F}_i(\mathbf{w}_i^{(t)}), \mathcal{F}_i(\epsilon_\theta(\mathcal{F}_i(\mathbf{w}_i^{(t)})))))$$

$$\text{Case 13} : \mathbf{w}_i^{(t-1)} = \psi^{(t)}(\mathbf{w}_i^{(t)}, \mathcal{F}_i(\phi^{(t)}(\mathbf{w}_i^{(t)}, \epsilon_\theta(\mathcal{F}_i(\mathbf{w}_i^{(t)})))))$$

$$\text{Case 14} : \mathbf{w}_i^{(t-1)} = \psi^{(t)}(\mathbf{w}_i^{(t)}, \mathcal{F}_i(\phi^{(t)}(\mathbf{w}_i^{(t)}, \mathcal{F}_i(\epsilon_\theta(\mathbf{w}_i^{(t)})))))$$

$$\text{Case 15} : \mathbf{w}_i^{(t-1)} = \psi^{(t)}(\mathbf{w}_i^{(t)}, \mathcal{F}_i(\phi^{(t)}(\mathbf{w}_i^{(t)}, \mathcal{F}_i(\epsilon_\theta(\mathcal{F}_i(\mathbf{w}_i^{(t)}))))))$$

$$\text{Case 16} : \mathbf{w}_i^{(t-1)} = \psi^{(t)}(\mathbf{w}_i^{(t)}, \mathcal{F}_i(\phi^{(t)}(\mathcal{F}_i(\mathbf{w}_i^{(t)}), \epsilon_\theta(\mathbf{w}_i^{(t)}))))$$

$$\text{Case 17} : \mathbf{w}_i^{(t-1)} = \psi^{(t)}(\mathbf{w}_i^{(t)}, \mathcal{F}_i(\phi^{(t)}(\mathcal{F}_i(\mathbf{w}_i^{(t)}), \epsilon_\theta(\mathcal{F}_i(\mathbf{w}_i^{(t)})))))$$

$$\text{Case 18} : \mathbf{w}_i^{(t-1)} = \psi^{(t)}(\mathbf{w}_i^{(t)}, \mathcal{F}_i(\phi^{(t)}(\mathcal{F}_i(\mathbf{w}_i^{(t)}), \mathcal{F}_i(\epsilon_\theta(\mathbf{w}_i^{(t)})))))$$

$$\text{Case 19} : \mathbf{w}_i^{(t-1)} = \psi^{(t)}(\mathbf{w}_i^{(t)}, \mathcal{F}_i(\phi^{(t)}(\mathcal{F}_i(\mathbf{w}_i^{(t)}), \mathcal{F}_i(\epsilon_\theta(\mathcal{F}_i(\mathbf{w}_i^{(t)}))))))$$

$$\text{Case 20} : \mathbf{w}_i^{(t-1)} = \psi^{(t)}(\mathcal{F}_i(\mathbf{w}_i^{(t)}), \phi^{(t)}(\mathbf{w}_i^{(t)}, \epsilon_\theta(\mathbf{w}_i^{(t)})))$$

$$\text{Case 21} : \mathbf{w}_i^{(t-1)} = \psi^{(t)}(\mathcal{F}_i(\mathbf{w}_i^{(t)}), \phi^{(t)}(\mathbf{w}_i^{(t)}, \epsilon_\theta(\mathcal{F}_i(\mathbf{w}_i^{(t)}))))$$

$$\text{Case 22} : \mathbf{w}_i^{(t-1)} = \psi^{(t)}(\mathcal{F}_i(\mathbf{w}_i^{(t)}), \phi^{(t)}(\mathbf{w}_i^{(t)}, \mathcal{F}_i(\epsilon_\theta(\mathbf{w}_i^{(t)}))))$$

$$\text{Case 23} : \mathbf{w}_i^{(t-1)} = \psi^{(t)}(\mathcal{F}_i(\mathbf{w}_i^{(t)}), \phi^{(t)}(\mathbf{w}_i^{(t)}, \mathcal{F}_i(\epsilon_\theta(\mathcal{F}_i(\mathbf{w}_i^{(t)})))))$$

$$\text{Case 24} : \mathbf{w}_i^{(t-1)} = \psi^{(t)}(\mathcal{F}_i(\mathbf{w}_i^{(t)}), \phi^{(t)}(\mathcal{F}_i(\mathbf{w}_i^{(t)}), \epsilon_\theta(\mathbf{w}_i^{(t)})))$$

$$\text{Case 25} : \mathbf{w}_i^{(t-1)} = \psi^{(t)}(\mathcal{F}_i(\mathbf{w}_i^{(t)}), \phi^{(t)}(\mathcal{F}_i(\mathbf{w}_i^{(t)}), \mathcal{F}_i(\epsilon_\theta(\mathbf{w}_i^{(t)}))))$$

$$\text{Case 26} : \mathbf{w}_i^{(t-1)} = \mathcal{F}_i(\psi^{(t)}(\mathbf{w}_i^{(t)}, \phi^{(t)}(\mathbf{w}_i^{(t)}, \mathcal{F}_i(\epsilon_\theta(\mathbf{w}_i^{(t)})))))$$

$$\text{Case 27} : \mathbf{w}_i^{(t-1)} = \psi^{(t)}(\mathcal{F}_i(\mathbf{w}_i^{(t)}), \mathcal{F}_i(\phi^{(t)}(\mathbf{w}_i^{(t)}, \epsilon_\theta(\mathbf{w}_i^{(t)}))))$$

$$\text{Case 28} : \mathbf{w}_i^{(t-1)} = \psi^{(t)}(\mathcal{F}_i(\mathbf{w}_i^{(t)}), \mathcal{F}_i(\phi^{(t)}(\mathbf{w}_i^{(t)}, \epsilon_\theta(\mathcal{F}_i(\mathbf{w}_i^{(t)})))))$$

$$\text{Case 29} : \mathbf{w}_i^{(t-1)} = \psi^{(t)}(\mathcal{F}_i(\mathbf{w}_i^{(t)}), \mathcal{F}_i(\phi^{(t)}(\mathbf{w}_i^{(t)}, \mathcal{F}_i(\epsilon_\theta(\mathbf{w}_i^{(t)})))))$$

Case 30 : $\mathbf{w}_i^{(t-1)} = \psi^{(t)}(\mathcal{F}_i(\mathbf{w}_i^{(t)}), \mathcal{F}_i(\phi^{(t)}(\mathbf{w}_i^{(t)}, \mathcal{F}_i(\epsilon_\theta(\mathcal{F}_i(\mathbf{w}_i^{(t)}))))))$

Case 31 : $\mathbf{w}_i^{(t-1)} = \psi^{(t)}(\mathcal{F}_i(\mathbf{w}_i^{(t)}), \mathcal{F}_i(\phi^{(t)}(\mathcal{F}_i(\mathbf{w}_i^{(t)}), \epsilon_\theta(\mathbf{w}_i^{(t)}))))$

Case 32 : $\mathbf{w}_i^{(t-1)} = \mathcal{F}_i(\psi^{(t)}(\mathbf{w}_i^{(t)}, \mathcal{F}_i(\phi^{(t)}(\mathbf{w}_i^{(t)}, \epsilon_\theta(\mathbf{w}_i^{(t)})))))$

Case 33 : $\mathbf{w}_i^{(t-1)} = \psi^{(t)}(\mathcal{F}_i(\mathbf{w}_i^{(t)}), \mathcal{F}_i(\phi^{(t)}(\mathcal{F}_i(\mathbf{w}_i^{(t)}), \mathcal{F}_i(\epsilon_\theta(\mathbf{w}_i^{(t)})))))$

Case 34 : $\mathbf{w}_i^{(t-1)} = \mathcal{F}_i(\psi^{(t)}(\mathbf{w}_i^{(t)}, \mathcal{F}_i(\phi^{(t)}(\mathbf{w}_i^{(t)}, \mathcal{F}_i(\epsilon_\theta(\mathbf{w}_i^{(t)}))))))$.

Similarly, four cases are derived from the trajectory 2 shown in part (c) of Figure 16. These correspond to Cases 35-38 below:

Case 35 : $\mathbf{w}_i^{(t-1)} = \psi^{(t)}(\mathbf{w}_i^{(t)}, f_i(\phi^{(t)}(\mathcal{A}(\{g_j(\mathbf{w}_j^{(t)})\}), \mathcal{A}(\{g_j(\epsilon_\theta(\mathbf{w}_j^{(t)}))\}))))$

Case 36 : $\mathbf{w}_i^{(t-1)} = \psi^{(t)}(\mathbf{w}_i^{(t)}, f_i(\phi^{(t)}(\mathcal{A}(\{g_j(\mathbf{w}_j^{(t)})\}), \mathcal{A}(\{g_j(\epsilon_\theta(\mathcal{F}_i(\mathbf{w}_j^{(t)})))\}))))$

Case 37 : $\mathbf{w}_i^{(t-1)} = \psi^{(t)}(\mathcal{F}_i(\mathbf{w}_i^{(t)}), f_i(\phi^{(t)}(\mathcal{A}(\{g_j(\mathbf{w}_j^{(t)})\}), \mathcal{A}(\{g_j(\epsilon_\theta(\mathbf{w}_j^{(t)}))\}))))$

Case 38 : $\mathbf{w}_i^{(t-1)} = \psi^{(t)}(\mathcal{F}_i(\mathbf{w}_i^{(t)}), f_i(\phi^{(t)}(\mathcal{A}(\{g_j(\mathbf{w}_j^{(t)})\}), \mathcal{A}(\{g_j(\epsilon_\theta(\mathcal{F}_i(\mathbf{w}_j^{(t)})))\}))))$.

The trajectory 3 shown in part (e) of Figure 16 accounts for two cases, corresponding to Cases 39-40 below:

Case 39 : $\mathbf{w}_i^{(t-1)} = f_i(\psi^{(t)}(\mathcal{A}(\{g_j(\mathbf{w}_j^{(t)})\}), \phi^{(t)}(\mathcal{A}(\{g_j(\mathbf{w}_j^{(t)})\}), \mathcal{A}(\{g_j(\epsilon_\theta(\mathbf{w}_j^{(t)}))\}))))$

Case 40 : $\mathbf{w}_i^{(t-1)} = f_i(\psi^{(t)}(\mathcal{A}(\{g_j(\mathbf{w}_j^{(t)})\}), \phi^{(t)}(\mathcal{A}(\{g_j(\mathbf{w}_j^{(t)})\}), \mathcal{A}(\{g_j(\epsilon_\theta(\mathcal{F}_i(\mathbf{w}_j^{(t)})))\}))))$.

Lastly, the trajectory 4 shown in part (g) of Figure 16 includes Cases 41-48 below:

Case 41 : $\mathbf{w}_i^{(t-1)} = f_i(\psi^{(t)}(\mathcal{A}(\{g_j(\mathbf{w}_j^{(t)})\}), \mathcal{A}(\{g_j(\phi^{(t)}(\mathbf{w}_j^{(t)}, \epsilon_\theta(\mathbf{w}_j^{(t)})))\})))$

Case 42 : $\mathbf{w}_i^{(t-1)} = f_i(\psi^{(t)}(\mathcal{A}(\{g_j(\mathbf{w}_j^{(t)})\}), \mathcal{A}(\{g_j(\phi^{(t)}(\mathbf{w}_j^{(t)}, \epsilon_\theta(\mathcal{F}_i(\mathbf{w}_j^{(t)}))))\})))$

Case 43 : $\mathbf{w}_i^{(t-1)} = f_i(\psi^{(t)}(\mathcal{A}(\{g_j(\mathbf{w}_j^{(t)})\}), \mathcal{A}(\{g_j(\phi^{(t)}(\mathbf{w}_j^{(t)}, \mathcal{F}_i(\epsilon_\theta(\mathbf{w}_j^{(t)}))))\})))$

Case 44 : $\mathbf{w}_i^{(t-1)} = f_i(\psi^{(t)}(\mathcal{A}(\{g_j(\mathbf{w}_j^{(t)})\}), \mathcal{A}(\{g_j(\phi^{(t)}(\mathbf{w}_j^{(t)}, \mathcal{F}_i(\epsilon_\theta(\mathcal{F}_i(\mathbf{w}_j^{(t)})))))\})))$

Case 45 : $\mathbf{w}_i^{(t-1)} = f_i(\psi^{(t)}(\mathcal{A}(\{g_j(\mathbf{w}_j^{(t)})\}), \mathcal{A}(\{g_j(\phi^{(t)}(\mathcal{F}_i(\mathbf{w}_j^{(t)}), \epsilon_\theta(\mathbf{w}_j^{(t)})))\})))$

Case 46 : $\mathbf{w}_i^{(t-1)} = f_i(\psi^{(t)}(\mathcal{A}(\{g_j(\mathbf{w}_j^{(t)})\}), \mathcal{A}(\{g_j(\phi^{(t)}(\mathcal{F}_i(\mathbf{w}_j^{(t)}), \epsilon_\theta(\mathcal{F}_i(\mathbf{w}_j^{(t)}))))\})))$

Case 47 : $\mathbf{w}_i^{(t-1)} = f_i(\psi^{(t)}(\mathcal{A}(\{g_j(\mathbf{w}_j^{(t)})\}), \mathcal{A}(\{g_j(\phi^{(t)}(\mathcal{F}_i(\mathbf{w}_j^{(t)}), \mathcal{F}_i(\epsilon_\theta(\mathbf{w}_j^{(t)}))))\})))$

Case 48 : $\mathbf{w}_i^{(t-1)} = f_i(\psi^{(t)}(\mathcal{A}(\{g_j(\mathbf{w}_j^{(t)})\}), \mathcal{A}(\{g_j(\phi^{(t)}(\mathcal{F}_i(\mathbf{w}_j^{(t)}), \mathcal{F}_i(\epsilon_\theta(\mathcal{F}_i(\mathbf{w}_j^{(t)})))))\})))$.

### H.3  Canonical Variable Denoising Process

Here, we present all possible canonical variable denoising processes. Due to the absence of noise prediction in the canonical space, a process first *redirects* canonical variable $\mathbf{z}^{(t)}$ to the instance spaces where a subsequence of operations $\epsilon_\theta(\cdot)$, $\phi^{(t)}(\cdot, \cdot)$ and $\psi^{(t)}(\cdot, \cdot)$ are computed.

We exclude the application of $\mathcal{F}_i$ to $\mathbf{w}_i^{(t)} \leftarrow f_i(\mathbf{z}^{(t)})$, as the variable remains unchanged after the operation. Therefore, applying $\mathcal{F}_i$ to $\mathbf{w}_i^{(t)} \leftarrow f_i(\mathbf{z}^{(t)})$ for the inputs of $\epsilon_\theta(\cdot)$, $\phi^{(t)}(\cdot, \cdot)$ and $\psi^{(t)}(\cdot, \cdot)$ is not considered.

Case 4 which belongs to the trajectory 3, is visualized in part (f) of Figure 16. Cases 5 and 49 derive from the trajectory 4 which are shown in part (h) of Figure 16.

Case 49 : $\mathbf{z}^{(t-1)} = \psi^{(t)}(\mathbf{z}^{(t)}, \mathcal{A}(\{g_i(\phi^{(t)}(f_i(\mathbf{z}^{(t)}), \mathcal{F}_i(\epsilon_\theta(f_i(\mathbf{z}^{(t)})))))\})))$

In the trajectory 1, $2^2 = 4$ cases are possible, as shown in part (b) of Figure 16. This includes Case 6 along with Cases 50-52 below:

Case 50 : $\mathbf{z}^{(t-1)} = \mathcal{A}(\{g_i(\psi^{(t)}(f_i(\mathbf{z}^{(t)}), \phi^{(t)}(f_i(\mathbf{z}^{(t)}), \mathcal{F}_i(\epsilon_\theta(f_i(\mathbf{z}^{(t)}))))))\})$

Case 51 : $\mathbf{z}^{(t-1)} = \mathcal{A}(\{g_i(\psi^{(t)}(f_i(\mathbf{z}^{(t)}), \mathcal{F}_i(\phi^{(t)}(f_i(\mathbf{z}^{(t)}), \epsilon_\theta(f_i(\mathbf{z}^{(t)}))))))\})$

Case 52 : $\mathbf{z}^{(t-1)} = \mathcal{A}(\{g_i(\psi^{(t)}(f_i(\mathbf{z}^{(t)}), \mathcal{F}_i(\phi^{(t)}(f_i(\mathbf{z}^{(t)}), \mathcal{F}_i(\epsilon_\theta(f_i(\mathbf{z}^{(t)}))))))\})$.

Lastly, trajectory 2, shown in part (d) of Figure 16, encompasses one possible case, corresponding to Case 53:

Case 53 : $\mathbf{z}^{(t-1)} = \mathcal{A}(\{g_i(\psi^{(t)}(f_i(\mathbf{z}^{(t)}), f_i(\phi^{(t)}(\mathbf{z}^{(t)}, \mathcal{A}(\{g_i(\epsilon_\theta(f_i(\mathbf{z}^{(t)})))\})))))\})$.

## H.4   Combined Variable Denoising Process

In this section, we introduce combined variable denoising processes where both instance and canonical variables are denoised. This process synchronizes instance variables and a canonical variable by aggregating the unprojected instance variables and the canonical variable in the canonical space.

For clarity, we introduce additional operations below. $\boldsymbol{\delta}_{\mathcal{Z}}(\cdot)$ takes a variable in the canonical space $\mathbf{z} \in \mathcal{Z}$, projects it into the instance spaces, predicts noises in those spaces, and aggregates them back in the canonical space after the unprojection. $\boldsymbol{\Phi}_{\mathcal{Z}}^{(t)}(\cdot)$ then computes Tweedie's formula [46] based on the noise term computed by $\boldsymbol{\delta}_{\mathcal{Z}}(\cdot)$.

$$\boldsymbol{\delta}_{\mathcal{Z}}(\mathbf{z}) = \mathcal{A}(\{g_i(\epsilon_\theta(f_i(\mathbf{z})))\}) \tag{30}$$

$$\boldsymbol{\Phi}_{\mathcal{Z}}^{(t)}(\mathbf{z}) = \phi^{(t)}(\mathbf{z}, \boldsymbol{\delta}(\mathbf{z})). \tag{31}$$

Similarly, given a set of variables in the instance spaces $\{\mathbf{w}_i\}$, the following operators aggregate the unprojected outputs of $\psi^{(t)}(\cdot, \cdot)$, $\epsilon_\theta(\cdot)$ and $\phi^{(t)}(\cdot, \cdot)$ in the canonical space:

$$\boldsymbol{\Psi}_{\mathcal{W}_i}^{(t)}(\{\mathbf{w}_i\}) = \mathcal{A}(\{g_i(\psi^{(t)}(\mathbf{w}_i^{(t)}, \phi^{(t)}(\mathbf{w}_i^{(t)}, \epsilon_\theta(\mathbf{w}_i^{(t)}))))\}) \tag{32}$$

$$\boldsymbol{\delta}_{\mathcal{W}_i}(\{\mathbf{w}_i\}) = \mathcal{A}(\{g_i(\epsilon_\theta(\mathbf{w}_i^{(t)}))\}) \tag{33}$$

$$\boldsymbol{\Phi}_{\mathcal{W}_i}^{(t)}(\{\mathbf{w}_i\}) = \mathcal{A}(\{g_i(\phi^{(t)}(\mathbf{w}_i^{(t)}, \epsilon_\theta(\mathbf{w}_i^{(t)})))\}) \tag{34}$$

We present joint variable denoising cases on the representative cases discussed in Section H:

Case 54 : $\mathbf{w}_i^{(t-1)} = \psi^{(t)}(\mathbf{w}_i^{(t)}, \phi^{(t)}(\mathbf{w}_i^{(t)}, f_i(\mathcal{A}(\{\boldsymbol{\delta}_{\mathcal{W}_i}(\{\mathbf{w}_i^{(t)}\}), \boldsymbol{\delta}_{\mathcal{Z}}(\mathbf{z}^{(t)})\}))))$

Case 55 : $\mathbf{w}_i^{(t-1)} = \psi^{(t)}(\mathbf{w}_i^{(t)}, f_i(\mathcal{A}(\{\boldsymbol{\Phi}_{\mathcal{W}_i}^{(t)}(\{\mathbf{w}_i^{(t)}\}), \boldsymbol{\Phi}_{\mathcal{Z}}^{(t)}(\mathbf{z}^{(t)})\})))$

Case 56 : $\mathbf{w}_i^{(t-1)} = f_i(\mathcal{A}(\{\boldsymbol{\Psi}_{\mathcal{W}_i}^{(t)}(\{\mathbf{w}_i^{(t)}\}), \mathbf{z}^{(t-1)}\}))$

Case 57 : $\mathbf{z}^{(t-1)} = \psi^{(t)}(\mathbf{z}^{(t)}, \phi^{(t)}(\mathbf{z}^{(t)}, \mathcal{A}(\{\boldsymbol{\delta}_{\mathcal{Z}}(\mathbf{z}^{(t)}), \boldsymbol{\delta}_{\mathcal{W}_i}(\{\mathbf{w}_i^{(t)}\})\})))$

Case 58 : $\mathbf{z}^{(t-1)} = \psi^{(t)}(\mathbf{z}^{(t)}, \mathcal{A}(\{\boldsymbol{\Phi}_{\mathcal{W}_i}^{(t)}(\{f_i(\mathbf{z}^{(t)})\}), \boldsymbol{\Phi}_{\mathcal{W}_i}^{(t)}(\{\mathbf{w}_i^{(t)}\})\})$

Case 59 : $\mathbf{z}_{t-1} = \mathcal{A}(\{\boldsymbol{\Psi}_{\mathcal{W}_i}^{(t)}(\{f_i(\mathbf{z}^{(t)})\}), \mathcal{A}(\{g_i(\mathbf{w}_i^{(t-1)})\})\})$.

Cases 54-59 correspond to the combined variable denoising processes from Cases 1-6, respectively. In each of the above cases, we highlight the terms already present in the original representative case in orange and newly added variable to be synchronized together in purple.

## H.5   Quantitative Results

In Table 11, we present the quantitative results of the 60 diffusion synchronization processes listed above. We follow the same toy experiment setup described in Section 3.4.1 and Section B.1. As outlined in Section H.1, for all instance variable denoising processes, including the independent denoising case (No Synchronization), we perform the final synchronization at the end of the denoising process. For $n$-to-1 projection, we utilize $M = 10$ multiplane images as done in Section 3.4.4.

We report the quantitative results of all cases in Table 11. The results align with the observations of Table 1. In the 1-to-1 projection scenario, most diffusion synchronization processes exhibit similar

performances. Except for Cases 55-56, the combined variable denoising processes (Cases 54-59) show suboptimal performances with FID [21] scores over 100. This indicates that denoising either instance variables or a canonical variable is sufficient to produce satisfactory and consistent results.

When it comes to the $1$-to-$n$ projection scenario, Cases 2 and 5 outperform the others, with some exceptions such as Cases 11 and 35. This trend is also consistent with the results in Section 3.4.3, highlighting the effectiveness of synchronizing the outputs of Tweedie's formula [46] $\phi^{(t)}(\cdot, \cdot)$ even when compared to more complex diffusion synchronization processes.

Lastly, in the $n$-to-$1$ projection scenario, Case 2 (SyncTweedies) is the only one that outperforms the others across all metrics.

In conclusion, as shown in Table 11, Case 2 (SyncTweedies) distinctly exhibits superior performance across various projection scenarios, outperforming even more convoluted cases.

Table 11: **A quantitative comparison of all cases in ambiguous image generation.** KID [6] is scaled by $10^3$. For each column, we highlight the row whose value is within 95% of the best.

| | CLIP-A [42] ↑ | CLIP-C [42] ↑ | FID [21] ↓ | KID [6] ↓ |
|---|---|---|---|---|
| | | 1-to-1 Projection | | |
| No Sync. | 28.49 | 62.0 | 102.14 | 51.72 |
| Case 1 | 30.26 | 64.45 | 85.55 | 31.95 |
| **Case 2 (SyncTweedies)** | 30.35 | 64.52 | 85.36 | 31.82 |
| Case 3 | 30.34 | 64.46 | 85.31 | 31.57 |
| Case 4 | 30.28 | 64.47 | 85.37 | 32.44 |
| Case 5 | 30.36 | 64.43 | 85.56 | 32.1 |
| Case 6 | 30.3 | 64.49 | 84.75 | 31.29 |
| Case 7 | 30.33 | 64.51 | 84.6 | 31.93 |
| Case 8 | 30.27 | 64.49 | 85.03 | 32.08 |
| Case 9 | 29.46 | 62.17 | 97.48 | 44.21 |
| Case 10 | 30.36 | 64.68 | 84.79 | 31.44 |
| Case 11 | 30.31 | 64.48 | 85.81 | 32.26 |
| Case 12 | 30.31 | 64.53 | 84.48 | 31.56 |
| Case 13 | 30.33 | 64.46 | 85.83 | 32.35 |
| Case 14 | 30.33 | 64.57 | 85.69 | 32.36 |
| Case 15 | 30.35 | 64.63 | 85.6 | 32.17 |
| Case 16 | 30.34 | 64.57 | 85.9 | 32.55 |
| Case 17 | 30.32 | 64.5 | 85.66 | 32.3 |
| Case 18 | 30.31 | 64.63 | 85.48 | 32.35 |
| Case 19 | 30.33 | 64.53 | 85.18 | 31.38 |
| Case 20 | 29.91 | 63.48 | 92.18 | 38.44 |
| Case 21 | 30.3 | 64.41 | 85.54 | 32.18 |
| Case 22 | 30.33 | 64.61 | 85.99 | 32.41 |
| Case 23 | 30.31 | 64.59 | 85.17 | 31.77 |
| Case 24 | 30.06 | 63.91 | 91.82 | 37.62 |
| Case 25 | 30.3 | 64.46 | 85.41 | 32.22 |
| Case 26 | 30.36 | 64.59 | 84.98 | 31.93 |
| Case 27 | 30.31 | 64.49 | 84.89 | 31.8 |
| Case 28 | 30.33 | 64.42 | 85.34 | 32.61 |
| Case 29 | 30.33 | 64.55 | 85.93 | 32.29 |
| Case 30 | 30.33 | 64.51 | 85.03 | 31.72 |
| Case 31 | 30.32 | 64.42 | 85.95 | 32.91 |
| Case 32 | 30.33 | 64.5 | 85.78 | 32.35 |
| Case 33 | 30.34 | 64.63 | 85.77 | 32.4 |
| Case 34 | 30.37 | 64.66 | 84.99 | 31.84 |
| Case 35 | 30.36 | 64.59 | 85.39 | 31.64 |
| Case 36 | 30.34 | 64.55 | 84.59 | 31.8 |
| Case 37 | 30.33 | 64.63 | 85.21 | 31.84 |
| Case 38 | 30.39 | 64.56 | 84.75 | 31.82 |
| Case 39 | 30.31 | 64.55 | 85.56 | 32.67 |
| Case 40 | 30.29 | 64.55 | 85.44 | 32.17 |
| Case 41 | 30.31 | 64.48 | 85.53 | 32.02 |
| Case 42 | 30.35 | 64.47 | 85.62 | 32.68 |
| Case 43 | 30.31 | 64.55 | 85.4 | 32.09 |
| Case 44 | 30.3 | 64.51 | 86.13 | 32.55 |
| Case 45 | 30.32 | 64.42 | 85.44 | 32.25 |
| Case 46 | 30.34 | 64.51 | 85.59 | 32.67 |
| Case 47 | 30.32 | 64.52 | 85.06 | 31.76 |
| Case 48 | 30.35 | 64.57 | 84.95 | 31.96 |
| Case 49 | 30.3 | 64.48 | 85.46 | 32.43 |
| Case 50 | 30.31 | 64.46 | 86.49 | 32.97 |
| Case 51 | 30.3 | 64.47 | 85.83 | 32.41 |
| Case 52 | 30.35 | 64.62 | 86.05 | 32.44 |
| Case 53 | 30.28 | 64.45 | 85.38 | 32.21 |

| | CLIP-A [42] ↑ | CLIP-C [42] ↑ | FID [21] ↓ | KID [6] ↓ |
|---|---|---|---|---|
| Case 54 | 21.27 | 50.03 | 422.05 | 491.69 |
| Case 55 | 30.09 | 64.26 | 87.04 | 34.26 |
| Case 56 | 30.33 | 64.4 | 87.26 | 33.06 |
| Case 57 | 21.28 | 49.99 | 420.29 | 495.9 |
| Case 58 | 27.02 | 60.08 | 113.68 | 57.55 |
| Case 59 | 26.88 | 60.05 | 116.9 | 60.24 |
| | | 1-to-$n$ Projection | | |
| No Sync. | 27.64 | 56.97 | 132.6 | 107.41 |
| Case 1 | 26.12 | 54.97 | 231.23 | 212.39 |
| **Case 2 (SyncTweedies)** | 30.23 | 60.87 | 110.73 | 76.25 |
| Case 3 | 29.96 | 60.49 | 118.53 | 86.38 |
| Case 4 | 25.86 | 54.29 | 254.74 | 250.76 |
| Case 5 | 30.29 | 61.0 | 108.77 | 74.51 |
| Case 6 | 30.0 | 60.55 | 117.64 | 84.84 |
| Case 7 | 28.41 | 58.41 | 195.37 | 186.39 |
| Case 8 | 25.79 | 54.21 | 271.32 | 278.01 |
| Case 9 | 28.77 | 57.56 | 129.27 | 99.92 |
| Case 10 | 30.11 | 60.91 | 115.29 | 84.28 |
| Case 11 | 30.2 | 60.84 | 110.38 | 76.53 |
| Case 12 | 29.92 | 60.8 | 121.93 | 86.54 |
| Case 13 | 29.93 | 60.84 | 121.64 | 86.13 |
| Case 14 | 29.3 | 59.91 | 163.26 | 127.25 |
| Case 15 | 29.53 | 60.29 | 158.83 | 140.3 |
| Case 16 | 30.05 | 60.45 | 117.48 | 86.86 |
| Case 17 | 30.1 | 60.74 | 118.56 | 88.96 |
| Case 18 | 30.18 | 60.77 | 113.1 | 79.59 |
| Case 19 | 30.17 | 60.79 | 117.68 | 85.37 |
| Case 20 | 29.45 | 59.41 | 119.8 | 87.59 |
| Case 21 | 30.06 | 60.76 | 115.39 | 82.83 |
| Case 22 | 30.02 | 60.58 | 116.08 | 83.42 |
| Case 23 | 30.06 | 60.69 | 117.55 | 84.91 |
| Case 24 | 29.45 | 59.43 | 121.43 | 89.89 |
| Case 25 | 29.94 | 60.45 | 118.35 | 86.34 |
| Case 26 | 30.02 | 60.55 | 119.02 | 86.96 |
| Case 27 | 29.92 | 60.45 | 118.86 | 86.83 |
| Case 28 | 30.01 | 60.52 | 119.1 | 86.91 |
| Case 29 | 29.94 | 60.54 | 118.76 | 85.98 |
| Case 30 | 29.97 | 60.52 | 120.49 | 89.38 |
| Case 31 | 29.95 | 60.45 | 119.19 | 86.53 |
| Case 32 | 30.02 | 60.62 | 119.03 | 86.73 |
| Case 33 | 29.95 | 60.52 | 119.92 | 87.55 |
| Case 34 | 29.96 | 60.54 | 119.57 | 87.29 |
| Case 35 | 30.2 | 60.84 | 110.38 | 76.08 |
| Case 36 | 29.92 | 60.8 | 121.93 | 86.99 |
| Case 37 | 29.94 | 60.45 | 118.35 | 86.27 |
| Case 38 | 30.02 | 60.55 | 119.02 | 86.99 |
| Case 39 | 29.94 | 60.45 | 118.35 | 86.05 |
| Case 40 | 30.02 | 60.55 | 119.02 | 87.07 |
| Case 41 | 29.92 | 60.45 | 118.86 | 86.51 |
| Case 42 | 30.01 | 60.52 | 119.1 | 86.46 |
| Case 43 | 29.94 | 60.54 | 118.76 | 86.42 |
| Case 44 | 29.97 | 60.52 | 120.49 | 89.04 |
| Case 45 | 29.95 | 60.45 | 119.19 | 86.38 |
| Case 46 | 30.02 | 60.62 | 119.03 | 87.21 |
| Case 47 | 29.95 | 60.52 | 119.92 | 87.91 |
| Case 48 | 29.96 | 60.54 | 119.57 | 87.2 |
| Case 49 | 29.32 | 59.98 | 162.41 | 126.53 |
| Case 50 | 30.0 | 60.62 | 118.17 | 85.47 |
| Case 51 | 29.96 | 60.53 | 119.6 | 87.85 |
| Case 52 | 30.02 | 60.62 | 119.79 | 87.78 |
| Case 53 | 30.0 | 60.62 | 118.17 | 85.66 |
| Case 54 | 21.25 | 49.99 | 442.91 | 538.91 |
| Case 55 | 25.99 | 55.43 | 223.48 | 200.81 |
| Case 56 | 25.9 | 55.22 | 229.84 | 210.01 |
| Case 57 | 21.25 | 50.02 | 423.48 | 501.66 |
| Case 58 | 28.21 | 58.01 | 151.01 | 122.97 |
| Case 59 | 28.05 | 57.96 | 152.2 | 123.95 |
| | | $n$-to-1 Projection | | |
| No Sync. | 28.26 | 61.75 | 111.11 | 57.24 |
| Case 1 | 21.28 | 49.94 | 405.82 | 496.98 |
| **Case 2 (SyncTweedies)** | 29.56 | 63.1 | 96.3 | 40.91 |
| Case 3 | 21.58 | 50.58 | 243.23 | 151.11 |
| Case 4 | 21.33 | 50.05 | 301.2 | 233.11 |
| Case 5 | 21.09 | 50.04 | 289.82 | 213.45 |
| Case 6 | 22.28 | 50.11 | 329.11 | 299.11 |
| Case 7 | 21.76 | 50.01 | 422.31 | 518.33 |

|         | CLIP-A [42] ↑ | CLIP-C [42] ↑ | FID [21] ↓ | KID [6] ↓ |
|---------|---------------|---------------|------------|-----------|
| Case 8  | 21.72 | 50.0  | 426.69 | 530.85 |
| Case 9  | 22.81 | 51.54 | 192.3  | 112.63 |
| Case 10 | 25.05 | 56.05 | 158.99 | 92.29  |
| Case 11 | 25.67 | 54.72 | 288.88 | 278.59 |
| Case 12 | 25.67 | 56.48 | 160.32 | 92.54  |
| Case 13 | 25.11 | 53.61 | 260.91 | 223.48 |
| Case 14 | 24.29 | 52.09 | 344.55 | 379.52 |
| Case 15 | 25.38 | 53.72 | 259.18 | 221.03 |
| Case 16 | 27.93 | 58.75 | 198.49 | 144.28 |
| Case 17 | 25.02 | 53.69 | 194.6  | 130.03 |
| Case 18 | 25.32 | 53.76 | 315.92 | 329.26 |
| Case 19 | 24.65 | 53.49 | 212.36 | 157.81 |
| Case 20 | 21.53 | 50.71 | 236.09 | 157.04 |
| Case 21 | 22.47 | 52.48 | 189.73 | 104.92 |
| Case 22 | 23.74 | 54.67 | 154.53 | 77.4   |
| Case 23 | 22.75 | 53.44 | 174.95 | 87.28  |
| Case 24 | 21.63 | 50.83 | 206.25 | 110.93 |
| Case 25 | 24.72 | 57.52 | 130.55 | 60.76  |
| Case 26 | 21.53 | 50.81 | 211.99 | 122.67 |
| Case 27 | 21.53 | 50.4  | 246.63 | 161.28 |
| Case 28 | 21.44 | 50.9  | 253.72 | 157.43 |
| Case 29 | 21.28 | 50.75 | 249.3  | 151.45 |
| Case 30 | 21.58 | 50.84 | 249.85 | 152.37 |
| Case 31 | 21.69 | 50.63 | 233.72 | 144.35 |
| Case 32 | 21.89 | 50.68 | 243.65 | 160.12 |
| Case 33 | 22.22 | 51.04 | 208.49 | 117.23 |
| Case 34 | 22.01 | 50.48 | 257.02 | 171.18 |
| Case 35 | 25.72 | 54.78 | 289.98 | 279.62 |
| Case 36 | 25.73 | 56.58 | 160.42 | 93.2   |
| Case 37 | 24.79 | 57.46 | 131.86 | 61.08  |
| Case 38 | 21.52 | 50.86 | 211.71 | 121.06 |
| Case 39 | 24.77 | 57.45 | 130.03 | 59.89  |
| Case 40 | 21.58 | 50.88 | 212.99 | 122.21 |
| Case 41 | 21.52 | 50.5  | 247.4  | 161.85 |
| Case 42 | 21.46 | 50.86 | 253.6  | 157.02 |
| Case 43 | 21.31 | 50.67 | 249.7  | 151.85 |
| Case 44 | 21.54 | 50.85 | 251.18 | 154.49 |
| Case 45 | 21.65 | 50.54 | 237.56 | 147.55 |
| Case 46 | 21.94 | 50.69 | 244.4  | 161.12 |
| Case 47 | 22.27 | 51.05 | 206.73 | 114.38 |
| Case 48 | 22.05 | 50.47 | 258.46 | 174.1  |
| Case 49 | 21.13 | 50.05 | 280.71 | 195.28 |
| Case 50 | 22.73 | 50.03 | 350.36 | 322.78 |
| Case 51 | 22.29 | 50.04 | 354.31 | 332.85 |
| Case 52 | 22.31 | 50.06 | 349.66 | 322.96 |
| Case 53 | 22.74 | 50.06 | 349.41 | 321.65 |
| Case 54 | 20.77 | 50.06 | 419.46 | 495.55 |
| Case 55 | 22.01 | 50.15 | 270.14 | 192.31 |
| Case 56 | 21.8  | 50.1  | 291.59 | 217.05 |
| Case 57 | 21.56 | 50.03 | 405.75 | 477.82 |
| Case 58 | 26.32 | 58.49 | 124.28 | 66.34  |
| Case 59 | 26.52 | 59.09 | 108.41 | 46.07  |

