# OpenReview forum: "SyncTweedies: A General Generative Framework Based on Synchronized Diffusions"
_NeurIPS.cc/2024/Conference — NeurIPS 2024 poster_

### Official Review · Reviewer_XFCg · 2024-07-07

**Soundness:** 3
**Presentation:** 4
**Contribution:** 3
**Rating:** 7
**Confidence:** 4

**Summary:**

SyncTweedie attempts to elucidate the design space for synchronized diffusion. Synchronized diffusion means to optimize some representation (e.g., 3D mesh) by jointly diffusing its lower-dimensional projections (e.g., 2D image). The paper analyzes existing synchronized diffusion approaches on different tasks under the same umbrella on a diagnostic benchmark, and provides an empirical insight that one underexplored design is the best choice. The paper then verifies the insight in a range of tasks, including panorama image generation and 3D texturing.

**Strengths:**

1. The paper is enjoyable to read. The analysis of the 5 categories of diffusion is very clear. The 3 types of tasks (1-1, 1-N, N-1) are also nicely defined. There have indeed been extensive studies on different tasks based on a similar idea, this paper has done a nice job of reviewing them and elucidating the design space.
2. The toy experiment is well-designed. It is a simple task, and the authors slightly modify it so it comprehensively covers all challenges in all types of tasks. The toy experiment is also diagnostic enough to provide empirical insights, which makes Case 2 (SyncTweedies) stands out.
3. The experiments cover a range of tasks including arbitrary-sized image generation, panorama image generation, mesh texturing, and 3DGS texturing. I find these tasks sufficient to support the generalizability of the proposed approach.

**Weaknesses:**

1. The depth-to-360$^\circ$ panorama result is blurry. In the meantime, the generated arbitrary-sized image seems to be fine. Is there any reason for that?
2. It would be better if generation diversity could be discussed, which I think would be an advantage over SDS-based works. How would different noise initialization affect the generated results and how robust is the approach given different random seeds?
3. Some implementation details are missing.
     - The paper seems to miss the implementation details of the aggregation function for different tasks. For example, how is the view aggregation for 3DGS performed? Is it a gradient-based optimization or an analytical weighted sum as in MultiDiffusion[4]? This will also help to understand the runtime of the method.
     - What are the number of projected views (i.e., $N$) for each task? How are the views sampled?

Minor:
- L95: tdeterministic -> deterministic
- L594 refers to something in L628.

**Questions:**

- The paper mentions that the framework cannot jointly optimize geometry and texture for 3D generation, could the authors elaborate more on why there is such a limitation? Does it suggest that the aggregation function $A$ cannot be too under-constrained?

**Limitations:**

Yes.

---

> ### Author Rebuttal · Authors · 2024-08-07
>
> Thank you for recognizing our work "enjoyable to read, and the analysis is very clear, elucidating the design space." We will correct the typos accordingly.
>
> Blurry outputs of depth-to-360-panorama.
> ---
>
> Depth-to-360-panorama involves Equirectangular projection which introduces more distortions than simple cropping in arbitrary-sized image generation task. We believe this can be mitigated using more precise mapping functions, which we leave for future work.
>
> Diversity of $\texttt{SyncTweedies}$.
> ---
> Thank you for pointing out the advantage that the diversity of $\texttt{SyncTweedies}$ can produce. In Figure R2 of **the attached PDF file**, we present qualitative results of 3D mesh texturing using the optimization-based method (Paint-it) and $\texttt{SyncTweedies}$ with different random seeds. $\texttt{SyncTweedies}$ generates more diverse texture images compared to Paint-it. We will add more results in the revised version.
>
> How is the view aggregation for 3DGS performed?
> ---
> In 3DGS texturing, optimization functions as a combination of unprojection and aggregation. As discussed in L661-663, we only optimize the colors of 3D Gaussians while fixing the other parameters, such as opacity, positions, and covariance matrices. The optimization runs for 2,000 iterations with a learning rate of 0.025. We will include the details of optimization in the revised version.
>
> How are the views sampled and what are the number of projected views $N$ for each task?
> ---
> In both $1$-to-$1$ and $1$-to-$n$ projections, the canonical space is represented as a single slab ($N=1$). For the $n$-to-$1$ projection experiment, however, the canonical space is represented by multiple slabs ($N=10$) to simulate the weighted sum rendering process.
>
> As outlined in L587-593 of Section A2.2, in 3D mesh texturing, we use eight views sampled around the object at $45^\circ$ azimuth intervals at $0^\circ$ elevation, along with two additional views at $0^\circ$ and $180^\circ$ azimuths with $30^\circ$ elevation. For 3DGS texturing, we use 50 views sampled from the Synthetic NeRF dataset, as described in L645-648 of Section A2.4.
>
> Why $\texttt{SyncTweedies}$ cannot optimize geometry and texture jointly?
> ---
> Our framework focuses on cases where projection and unprojection mappings are defined between the canonical space and the instance spaces. When the geometry is not given, the mappings cannot be defined, limiting the application of our method. In future work, initiating the generation process with a coarse geometry provided by an off-the-shelf network and jointly updating the geometry and appearances would be an interesting direction.

---

> > ### Comment · Reviewer_XFCg · 2024-08-11
> >
> > Thanks for the rebuttal. It addresses most of my concerns. The clarification on blurry output, implementation details, and diversity looks good to me. One minor point about the joint texture and geometry optimization: "When the geometry is not given, the mappings cannot be defined", I think the mapping could be the 3D-to-2D rendering. I will keep my score.

---

### Official Review · Reviewer_TnCt · 2024-07-11

**Soundness:** 3
**Presentation:** 2
**Contribution:** 3
**Rating:** 6
**Confidence:** 3

**Summary:**

This paper introduces SyncTweedies, a novel framework to generate diverse visual content such as ambiguous images, panoramas, mesh textures, and Gaussian splat textures. The method uses a synchronization process that averages outputs of Tweedie's formula across multiple instance spaces, eliminating the need for fine-tuning. The authors claim their method preserves the rich priors from large-scale datasets, enhancing generalizability. Experimental results show that SyncTweedies outperforms existing methods in various applications.

**Strengths:**

(1) From my point of view, the framework's versatility is impressive, as it addresses various visual content generation tasks. This general applicability is a significant advantage over more specialized methods.  The use of Tweedie's formula for synchronizing diffusion processes across different instance spaces is genuinely novel. This methodological innovation could inspire new directions in the field of generative models.

(2) SyncTweedies' ability to function without fine-tuning is a strong point. This feature ensures that the model retains its generalization capabilities, making it effective on diverse and previously unseen datasets.

(3) Besides, the authors have conducted extensive experiments, comparing their method against several state-of-the-art techniques. These comparisons convincingly demonstrate the superior performance of SyncTweedies across multiple tasks.

**Weaknesses:**

(1) The method assumes that the projection and unprojection functions are accurate, but in practice, these operations can introduce errors, especially in complex transformations such as those required for 3D mesh texturing. The author does not address how SyncTweedies handles these projection errors, which can accumulate and degrade the quality of the generated content.

(2) The theoretical basis of SyncTweedies seems to implicitly assume that the data distributions in the instance and canonical spaces are well-aligned. However, this assumption may not hold in practice, especially when dealing with diverse and complex datasets. The paper does not explore the theoretical implications of misaligned data distributions and how they might affect the performance and stability of the synchronization process.

**Questions:**

(1) How does SyncTweedies handle scenarios where projection and unprojection functions between instance spaces and the canonical space are not perfectly invertible? What impact does this have on the quality of generated content?

(2) I am also wondering if the synchronization process can be optimized further to reduce computational overhead without compromising the quality of the generated content.

**Limitations:**

1. The computational complexity and scalability of SyncTweedies are not thoroughly discussed (Lines 271-290). The synchronization process involves multiple denoising steps and averaging operations, which can be computationally intensive. There is no detailed analysis of the time and space complexity of the method, making it challenging to evaluate its feasibility for large-scale or real-time applications.

2. The paper assumes a level of consistency and stability across instance spaces, which may not always be realistic (Lines 202-214). For example, in 3D mesh texturing, different views can have significantly different appearances due to factors like lighting, occlusion, and perspective distortion. SyncTweedies does not provide a thorough analysis of how it handles such variability. Robustness under varying conditions is crucial for practical applications

---

> ### Author Rebuttal · Authors · 2024-08-07
>
> Thank you for your recognition of our work such as "the framework's versatility is impressive", and "the methodological innovation could inspire new directions in the field of generative models." Below, we answer your insightful comments.
>
> How does $\texttt{SyncTweedies}$ handle scenarios where mappings are not invertible?
> ---
> We would like to emphasize that one of our contributions is the analysis of diffusion synchronization cases with various mapping functions: $1$-to-$1$, $1$-to-$n$, and $n$-to-$1$ projections (Section 3.4.1, Section 3.4.2, and 3.4.3). The $1$-to-$n$ and $n$-to-$1$ projections correspond to cases where the mappings are not invertible. Our comprehensive analysis demonstrates that $\texttt{SyncTweedies}$ outperforms other methods across all mapping scenarios, even when the mappings are not invertible. Additionally, we provide results of applications corresponding to each mapping scenario and their analysis: $1$-to-$1$ projection (arbitrary-sized image generation, Section A3), $1$-to-$n$ projection (3D mesh texturing, Section 5.1 and depth-to-360-panorama, Section 5.2), and $n$-to-$1$ projection (3DGS texturing, Section 5.3).
>
> Assumption of aligned distributions between canonical and instance spaces.
> ---
> Thank you for providing insightful feedback. As the reviewer has mentioned, we agree that there might be a misalignment of data distributions between instance and canonical spaces. For example, in 3D mesh texturing, the data distribution of the pretrained image diffusion model in the instance space is more diverse than the distribution of rendered 3D mesh images. Even with potential distribution misalignment, we demonstrate that $\texttt{SyncTweedies}$ can be applied to diverse applications, outperforming both optimization-based and finetuning-based approaches. In future work, we plan to provide a theoretical analysis of misaligned data distributions.
>
> Optimizing synchronization process to reduce computational overhead without compromising the quality.
> ---
> While the synchronization process in $1$-to-$1$ and $1$-to-$n$ projection cases introduces some computational overhead, the majority of the computational overhead comes from predicting noise for multiple instance space samples. Similar to previous work such as MVDream and MVDiffusion, in applications where the canonical space sample is expressed as a composition of instance space samples, noise prediction for multiple instance space samples is inevitable. One possible direction to reduce computational overhead would be to use fewer views without sacrificing output quality, which we leave for future work.
>
> Time and space complexity of $\texttt{SyncTweedies}$.
> ---
> Please see Section A5 in the appendix for a time complexity comparison, where $\texttt{SyncTweedies}$ demonstrates an 8-11 times faster running time compared to optimization-based methods and a comparable or 3-7 times faster running time compared to iterative-view-update-based methods. In Table R1 of **the attached PDF file**, we include results comparing the space complexity of other baselines and $\texttt{SyncTweedies}$, where our method takes around 6-9 GiB of memory, making it suitable for most GPUs. We use NVIDIA RTX A6000 for computation comparisons.
>
> Analysis of how $\texttt{SyncTweedies}$ handles variability across views.
> ---
> We agree that introducing variability across views is an important factor. We demonstrate that our method can control variations across views by using varying prompts. In Figure R1 of **the attached PDF file**, we provide qualitative results of arbitrary-sized images generation using a uniform prompt and varying multiple prompts across views. The result shows that our method can generate canonical space samples with variations across views.

---

> > ### Comment · Reviewer_TnCt · 2024-08-11
> >
> > Thanks for submitting the rebuttal which addresses my concerns. I think this is a good paper and deserves publication. As such, I will increase my rating to weakly accept.

---

### Official Review · Reviewer_KJxR · 2024-07-11

**Soundness:** 3
**Presentation:** 3
**Contribution:** 3
**Rating:** 7
**Confidence:** 4

**Summary:**

This paper investigates content generation within a target space using pretrained diffusion models operating in projected subspaces. The authors analyze five variants of the DDIM procedure, performed separately in each subspace and aggregated in the target space using known projection and unprojection operators. They explore different sequences of projection and aggregation, concluding that the optimal approach is to aggregate each subspace's "estimated x0 from xt" (the output of Tweedie's formula).

The method is evaluated through texturing tasks involving depth, mesh, and 3DGS, each representing distinct projection and unprojection scenarios. The ablation study and comparisons against baselines demonstrate the robustness of the proposed aggregation stage.

**Strengths:**

+ The paper effectively breaks down the multi-DDIM process, clearly delineating distinct alternatives that differ in the sequencing of the projection and aggregation operations.
+ The identification of three different tasks, which span a range of projection and un-projection operations, enriches the analysis and allows for a clear identification of the strengths and weaknesses of each aggregation strategy. I also appreciated the toy experiments, which effectively mimic the 3D effect in a simpler 2D setup.
+ The final strategy advocated for in this work aligns with the standard DDIM, is straightforward to implement, requires no additional tuning, and does not incur extra computational costs.

**Weaknesses:**

- Missing Explanations: The figures are difficult to understand due to missing information. For example, Figure 3 does not explicitly state the experimental setup, including the definitions of the canonical space and the subspaces.
- The formulation of cases 1-3 that operate in the subspace does not explain how the recovered signal in z is finally returned. This is related to my previous comment, where it seems Figure 3 shows individual subspace generations rather than the end goal/task.
- The initialization process is not discussed in the main paper. It is unclear whether the subspaces share the same initial noise. If they do, how is noise being generated in z? This is particularly challenging when z is 3DGS.
- In the appendix, it states that the 3DGS are optimized towards the predicted clean images. Is this optimization considered the unprojection operation "g"? If so, this should be explicitly stated and moved to the main paper. More details on this optimization process are needed, including whether it is performed until convergence at each denoising time during generation.
- In lines 129-130, phi and psi are used in the target domain. Please explain how this is done properly, given that noise is not injected in that space. Why would the coefficients in eq (3) and (4) hold true?

- The caption of Figure 1, “Diverse visual content generated by SyncTweedies,” and the abstract phrase "generating diverse visual content" are misleading. They suggest the generation of general 3D content, while the method, as mentioned in the limitations, relies on known mappings between the subspaces and the target space, limiting it to texturing tasks. The authors should tone down these statements.
- If the mapping from the canonical space to the instance spaces is simple (say, linear or even the Idnetity mapping), it seems all "cases" perform similarly both quantitatively and qualitatively. Do these methods mathematically converge to the same method, or are there still differences? In the 1-1 projection example in Figure 3, it appears all methods produce the same result. Is this correct? Table 10A shows almost identical results for different methods, and in Table 1, the scores for the 1-1 cases are very similar.
- The claim of being the first to propose this method and that it was previously "overlooked" is too bold. Guidance literature shows that guidance in "\hat{x_0}" offers more stable optimization and results. While guidance differs from multi-diffusion, it is similar enough to make the analogy. The authors should discuss similarities and differences, referencing papers like "Universal Guidance for Diffusion Models" and motion diffusion papers such as MDM and Trace&Pace. Further, if in degenerate cases of f and g maps (e.g. if they are the identity function) some "cases" are indistinguishable, be mindful when attributing aggregation choices to these methods (e.g., MultiDiffusion). Methods requiring tuning, like DiffCollage, should not be dismissed as baselines. The main contribution of this paper is identifying the best "place" to aggregate, and tuning is somewhat orthogonal. If "Case 2" improves DiffCollage, this should be demonstrated.
- The term "1:1 case" is confusing and suggests disjoint individual subspaces.
- The integration process of 1:n should also be better explained. It may also be useful to add illustration figures.
- Improve Figure 2 for clarity with a more detailed caption. Connecting the diagram to the notations introduced in the paper would also help readability.

**Questions:**

please see weaknesses.

**Limitations:**

See weaknesses -- the work in it sucrrent form is limited to known geometry

---

> ### Author Rebuttal · Authors · 2024-08-07
>
> Thank you for your in-depth reviews and feedback. We find that some of the reviews have been addressed in the appendix but are missing in the main paper. Please understand that these details were moved to the appendix due to page limits. However, we agree that providing more details in the main paper will help in understanding our work, and we will move these details to the main paper for clarity.
>
> **Please refer to the global rebuttal for our answers to "convergence to the same method in $1$-to-$1$ projection" and "comparison with finetuning-based methods."**
>
> Figure 3 experimental setup.
> ---
> Due to page limits, we detailed the experimental setup of ambiguous image generation in L559-583 of the appendix. Ambiguous image generation is a special case where both canonical and instance spaces take the form of images with the same dimensionality. Here, we consider one image as the canonical space, and instance space samples are obtained by applying transformations to the canonical space sample. We acknowledge that Figure 3 was not presented with sufficient details. We will move the experimental setup details from the appendix to the main paper in the final version.
>
> How $\mathbf{z}$ is returned in Cases 1-3?
> ---
> As discussed in L529-531, instance variable denoising processes, including Cases 1-3, obtain the final canonical sample $\mathbf{z}^{(0)}$ by aggregating fully denoised instance samples ${\mathbf{w}^{(0)}_i}$ in RGB space.
>
> Details on noise initialization.
> ---
> The details of the initialization scheme are provided in L517-521. For all experiments except 3DGS texturing, instance space samples are initialized by projecting the initial canonical space variable to each view, $\mathbf{w}_i = f_i(\mathbf{z}^{(T)})$, where $\mathbf{z}^{(T)} \sim \mathcal{N}(\mathbf{0}, \mathbf{I})$. For 3DGS texturing, instance variable denoising cases initialize noises directly in the instance space to prevent the variance decrease issue discussed in the paper. On the other hand, canonical denoising cases have no other option but to initialize noise in the canonical space $\mathbf{z}^{(T)} \sim \mathcal{N}(\mathbf{0}, \mathbf{I})$, which causes a variance decrease issue when projecting to the instance space. We will move the details from the appendix to the main text.
>
> In 3DGS, is optimization considered as the unprojection operation?
> ---
> Yes, more precisely, a combination of unprojection and aggregation functionsby optimization as outlined in L580-581. As discussed in L661-663, we only optimize the colors of 3D Gaussians while fixing the other parameters, such as opacity, positions, and covariance matrices. The optimization runs for 2,000 iterations with a learning rate of 0.025. We will include this further detail in the paper.
>
> How Equations (3) and (4) can be used in the target space? Why would the coefficients in Equations (3) and (4) hold true?
> ---
> As shown in the "convergence to the same method in $1$-to-$1$ projection" paragraph in the global rebuttal, when the mappings are $1$-to-$1$, all cases become identical, and canonical variable denoising processes follow the exact same trajectory as the instance variable denoising processes. Hence the coefficients defined by the instance space diffusion model can also be used in Cases 4-5. However, as discussed in Sections 3.4.2 and 3.4.3, the same may not hold when the mappings are $1$-to-$n$, as this may violate Equations R1 and R2 in the global rebuttal. Additionally, $n$-to-$1$ mappings result in a variance decrease when projecting the canonical space sample, causing performance degradation in canonical variable denoising cases. For clarity, we will add these details in our main paper.
>
> Figure 1 caption.
> ---
> The phrase "diverse visual content" refers to the ambiguous images, arbitrary-sized images, depth-to-360-panoramas, and textures on mesh and 3DGS that SyncTweedies produces, as shown in Figures 1, 3, and A7. We will adjust the phrase accordingly to better convey the capabilities and limitations of ours.
>
> First to propose this method and that it was previously overlooked is too bold.
> ---
> The previous works mentioned by the reviewer also utilize $\hat{\mathbf{x}}^{(0)}$, similar to $\texttt{SyncTweedies}$. However, we would like to clarify that the phrases "first to propose" (L40, L62) and "unexplored approach" (L64) in the main paper are meant to claim the first work to present a generalized diffusion synchronization framework and instance variable denoising process that averages the outputs of Tweedie's formula, not the first work utilizing the outputs of Tweedie's formula $\hat{\mathbf{x}}^{(0)}$. Nevertheless, we find that the related works utilizing $\hat{\mathbf{x}}^{(0)}$ will make our paper more comprehensive, and we will revise our related work accordingly.
>
> The term "1:1 case" is confusing and suggests disjoint individual subspaces.
> ---
> Please note that the instance spaces are not disjoint spaces. The instance spaces are initialized with overlapping regions, and the term $1$-to-$1$ is used to indicate the property of one canonical space pixel mapping to one instance space pixel. We acknowledge that the term $1$-to-$1$ may sound confusing, and we will consider revising the term to better indicate the property such as pixel-aligned mapping.
>
> The integration process of 1:n should also be better explained.
> ---
> We will add more details to the aggregation process of $1$-to-$n$ projection with an illustration figure. In $1$-to-$n$ projection, each instance space sample is unprojected to the canonical space resulting $N$ unprojected samples, $\\{g_i(\mathbf{w}^{(t)}) \\} _{i=1}^N $. The canonical space sample $\mathbf{z}^{(t)}$ is then obtained by aggregating the unprojected samples using a weighted sum, where the weights can be either a scalar $1/N$ or determined based on the visibility from each view $\\{ w_i \\} _{i=1}^N$.
>
> Figure 2 caption.
> ---
> We will add more details to the caption of Figure 2 with explanations of the notations.

---

> > ### Comment · Reviewer_KJxR · 2024-08-12
> >
> > I appreciate the authors detailed response. My concerns have been addressed and I recommend acceptance.

---

### Author Rebuttal · Authors · 2024-08-07

We appreciate all reviewers for their constructive and insightful comments. Some comments are addressed in the global rebuttal due to length constraints.

***Please refer to the attached PDF file for qualitative and quantitative results.***

[KJxR] Do cases mathematically converge to the same method when the mappings are $1$-to-$1$ projection?
---
Yes, it is mathematically guaranteed that Case 1-5 become identical when the mappings are $1$-to-$1$ and noises are initialized by projecting from the canonical space $\mathbf{w}_i^{(T)} = f_i(\mathbf{z}^{(T)})$, where $\mathbf{z}^{(T)} \sim \mathcal{N}(\mathbf{0}, \mathbf{I})$. Note that $\phi^{(t)}(\cdot, \cdot)$ and $\psi^{(t)}(\cdot, \cdot)$ are linear operations and commutative with other linear operation such as $f_i, \mathcal{A}$, and $g_i$. Assume the following conditions hold:

\\begin{align*}
    \\mathbf{z}^{(T)} &= \\mathcal{A}( \\{g_i( f_i(\\mathbf{z}^{(T)}))\\}), && \\text{(R1)} \\\\
    \\mathcal{A}(\\{g_i(\\mathbf{w}_i)\\}) &= \\mathcal{A}(\\{ g_i(f_i(\\mathcal{A}(\\{ g_j(\\mathbf{w}_j) \\}))) \\}) \\quad \\forall \\{\\mathbf{w}\\} _{i=1}^N. && \\text{(R2)}
\\end{align*}

Based on induction, we have:
\\begin{align*}
    \\mathbf{z}^{(t-1)}
    &= \\psi^{(t)}(\\mathbf{z}^{(t)}, \\phi^{(t)}(\\mathbf{z}^{(t)}, \\mathcal{A}( \\{ g_i(\\epsilon_\\theta(f_i((\\mathbf{z}^{(t)})))) \\}))) && \\text{(Case 4)}\\\\
    &= \\psi^{(t)}(\\mathbf{z}^{(t)}, \\phi^{(t)}(\\mathcal{A}( \\{ g_i( f_i(\\mathbf{z}^{(t)})) \\}), \\mathcal{A}( \\{ g_i(\\epsilon_\\theta(f_i(\\mathbf{z}^{(t)}))) \\}))) && \\\\
    &= \\psi^{(t)}(\\mathbf{z}^{(t)}, \\mathcal{A}( \\{ g_i(\\phi^{(t)}(f_i(\\mathbf{z}^{(t)}), \\epsilon_\\theta( f_i (\\mathbf{z}^{(t)})))) \\})) && \\text{(Case 5)}\\\\
    &= \\psi^{(t)}(\\mathcal{A}( \\{g_i( f_i(\\mathbf{z}^{(t)})) \\}), \\mathcal{A}(\\{ g_i( \\phi^{(t)}( f_i(\\mathbf{z}^{(t)}), \\epsilon_\\theta(f_i(\\mathbf{z}^{(t)})))) \\} )) && \\\\
    &= \\mathcal{A}( \\{ g_i(\\psi^{(t)}(f_i(\\mathbf{z}^{(t)}), \\phi^{(t)}( f_i(\\mathbf{z}^{(t)}), \\epsilon_\\theta(f_i(\\mathbf{z}^{(t)}))))) \\}) \\\\
    &= \\mathcal{A}(\\{ g_i( f_i( \\mathcal{A}(\\{ g_j(\\psi^{(t)}(f_j(\\mathbf{z}^{(t)}), \\phi^{(t)}(f_j(\\mathbf{z}^{(t)}), \\epsilon_\\theta(f_j(\\mathbf{z}^{(t)}))))) \\})))\\}) \\\\
    &= \\mathcal{A}(\\{ g_i( f_i(\\mathbf{z}^{(t-1)}))\\}),
\\end{align*}
where the last equality holds the induction hypothesis. This proves that Cases 4-5 are identical.
For instance variable denoising cases we have:
\\begin{align}
\\mathbf{w}^{(t-1)}_i &= \\psi^{(t)}(\\mathbf{w}^{(t)}_i, \\phi^{(t)}(\\mathbf{w} ^{(t)}_i, f_i( \\mathcal{A}( \\{g_j(\\epsilon _\\theta(\\mathbf{w}^{(t)}_j))\\} )))) && \\text{(Case 1)}\\\\
&= \\psi^{(t)}(\\mathbf{w}^{(t)}_i, \\phi^{(t)}( f_i( \\mathcal{A}( \\{ g_j( \\mathbf{w}^{(t)}_j) \\})), f_i( \\mathcal{A}( \\{ g_j( \\epsilon _\\theta(\\mathbf{w}^{(t)}_j)) \\} )))) && \\\\
&= \\psi^{(t)}(\\mathbf{w}^{(t)}_i, f_i( \\mathcal{A}( \\{ g_j(\\phi^{(t)}(\\mathbf{w}^{(t)}_j, \\epsilon _\\theta(\\mathbf{w}^{(t)}_j))) \\} ))) && \\text{(Case 2)}\\\\
&= \\psi^{(t)}( f_i( \\mathcal{A}( \\{ g_j( \\mathbf{w}^{(t)}_j ) \\})), f_i( \\mathcal{A}( \\{ g_j(\\phi^{(t)}(\\mathbf{w}^{(t)}_j, \\epsilon _\\theta(\\mathbf{w}^{(t)}_j))) \\}))) && \\\\
&= f_i( \\mathcal{A}( \\{ g_j(\\psi^{(t)}(\\mathbf{w}^{(t)}_j, \\phi^{(t)}(\\mathbf{w}^{(t)}_j, \\epsilon _\\theta(\\mathbf{w}^{(t)}_j)))) \\} )) && \\text{(Case 3)} \\\\
&= f_i( \\mathcal{A}( \\{ g_j( f_j( \\mathcal{A}( \\{ g_k (\\psi^{(t)}(\\mathbf{w}^{(t)}_k, \\phi^{(t)}(\\mathbf{w}^{(t)}_k, \\epsilon _\\theta(\\mathbf{w}^{(t)}_k)))) \\}))) \\})) \\\\
&= f_i( \\mathcal{A}( \\{ g_j(\\mathbf{w}_j^{(t-1)}) \\})),
\\end{align}
where the last equality holds the induction hypothesis. This proves that Cases 1-3 are identical.

We validate our observations both qualitatively and quantitatively in our $1$-to-$1$ projection toy experiment and arbitrary-sized image generation application in Section 3.4 and Section A3, respectively, where all cases generate identical results. However, the same may not hold when the mappings are $1$-to-$n$ or $n$-to-$1$. We will add these details in the revised version.

[KJxR] Methods requiring tuning should not be dismissed as baselines.
---
Even though a comparison with DiffCollage is not included since the code is not publicly released, we provide a comparison with other finetuning-based methods, such as Paint3D in 3D mesh texturing (Table 3) and MVDiffusion in depth-to-360-panorama generation tasks (Table 4) in the main paper. As discussed in L286-288 and L300-301 of the paper, $\texttt{SyncTweedies}$ outperforms the finetuned models since they are trained on a relatively small-scale dataset compared to the pretrained image diffusion models, which limits their applicability to unseen scenes. Additionally, based on the reviewer's comment, we experimented applying $\texttt{SyncTweedies}$ on a finetuned model, MVDiffusion. Qualitative results are presented in Figure R3 of **the attached PDF file**. We observe that a finetuned model with $\texttt{SyncTweedies}$ synchronization still fails to generalize to unseen domains, resulting in consistently inferior performance.

---

### Author Response · Authors · 2024-08-13

We sincerely thank the reviewers for taking the time and effort to review our paper. We appreciate thoughtful and constructive comments, and they are indeed invaluable in improving our work. We will address additional suggestions in our final revision.

---

### Decision · Program_Chairs · 2024-09-25

**Decision:**

Accept (poster)

**Comment:**

All reviewers recognized the quality of the work :
- "The identification of three different tasks [...] enriches the analysis and allows for a clear identification of the strengths and weaknesses of each aggregation strategy."
- "the framework's versatility is impressive"
- "This methodological innovation could inspire new directions in the field of generative models."
- "extensive experiments"
- "The paper is enjoyable to read"
- "The toy experiment is well-designed."
- "The experiments cover a range of tasks..."

Several concerns have been raised by the reviewers but the discussion period has resolved those and all reviewers have positively judged the authors response.